

# Fundamentals of Data Assimilation

Peter Rayner[1], Anna M. Michalak[2], and Frédéric Chevallier[3]

[1]School of Earth Sciences, University of Melbourne, Melbourne, Australia
[2]Dept. of Global Ecology, Carnegie Institution for Science, Stanford, USA
[3]Laboratoire des Sciences du Climat et de l'Environnement, Gif sur Yvette, France

*Correspondence to:* Peter Rayner (prayner@unimelb.edu.au)

**Abstract.** This article lays out the fundamentals of data assimilation as used in biogeochemistry. It demonstrates that all of the methods in widespread use within the field are special cases of the underlying Bayesian formalism. Methods differ in the assumptions they make and information they provide on the probability distributions used in Bayesian calculations. It thus provides a basis for comparison and choice among these methods. It also provides a standardised notation for the various quantities used in the field.

## 1  Introduction

The task of improving current knowledge by the consideration of observed phenomena is a fundamental part of the scientific process. The mechanics of the process are a matter for historians of science and a matter of fierce debate while its fundamental reliability is a matter for philosophers; literally a meta-physical question. Here we are concerned with algorithms for automating the process. The step by step construction of such algorithms from agreed upon logical premises is beyond our scope here but is masterfully presented by Jaynes and Bretthorst (2003).

It is important to note at the outset the generality of the machinery we will present, especially since many of the common objections to the methods spring from misunderstandings of this generality. For example it is a common complaint that data assimilation can improve a model but cannot test it. We will show in a later paper that the methodology easily extends to choice among an ensemble of models.

At the outset we should settle some questions of terminology. The phrases "data assimilation", "parameter estimation", "inverse modelling" and "model-data fusion" are used with overlapping meanings in the literature. Here we will draw no distinction between them. We are concerned with algorithms to improve the performance of models by the use of data. The improvement will come via knowledge of the state (quantities the model calculates), boundary conditions (quantities we insert into the model) or structure of the model. Performance will be assessed by the model's ability to match observations.

The application of data assimilation to biogeochemistry has proceeded from informality to formal methods and from small to larger problems. One can see the study of Fung et al. (1987) as a data assimilation experiment. These authors adjusted respiration parameters in a simple biosphere model so that the seasonal cycle of atmospheric $CO_2$ concentration matched better the observed seasonal cycle at some stations. Fifteen years later Kaminski et al. (2002) performed a similar study on





a similar model with more stations and a formal algorithm. We see similar developments in the estimation of $CO_2$ fluxes (e.g. Tans et al. (1990) vs Chevallier et al. (2010)) and ocean productivity (Balkanski et al. (1999) cf Brasseur et al. (2009))

The flourishing of the general approach has also led to a great diversification of methods, reflecting repeated borrowings from other fields. This has been extremely fruitful, but is confusing for a novice. Our aim in this paper is to reintroduce the fundamental theory and demonstrate how these many methods are its implementations. We will aim to tread a careful path between generality and simplicity. If we succeed, a reader should be well-placed to understand the relationships among the methods and applications presented later. We will also introduce a consistent notation general enough to capture the range of applications.

The structure of the paper is as follows: In Section 2 we introduce the general theory. There is no new material here, but we need a basis for the subsequent discussion. Section 3 introduces a consistent notation. Section 4 presents an overall approach to the solution to the data assimilation problem through the use of a simple example. Section 5 describes the construction of inputs to the problem. Section 6 describes solution methods and how they are special cases of the general theory which take advantage of the circumstances of a particular problem. Section 7 describes some computational aspects related to the solution of data assimilation problems.

## 2   Data Assimilation as Statistical Inference

### 2.1   Events and Probability

In what follows we will not include mathematically precise language. We think the trade-off of ambiguity for simplicity suits our introductory task. Readers are referred to Evans and Stark (2002), Jaynes and Bretthorst (2003) or Tarantola (2004) for more complete descriptions.

For our purposes we define an event as a statement about the condition of a system. This is deliberately general; such statements can take many forms. Examples include categorical or discrete statements (e.g. "including isopycnal mixing improves ocean circulation models"), logical propositions (e.g. "increasing soil temperatures lead to increased soil respiration") or quantitative statements like "the Amazon forest is a sink of between 1 and 2 $pgCy^{-1}$". The choice of the set of events we consider is the first one we make setting up any data assimilation problem. We require that any events which are logically incompatible are disjoint (mutually exclusive) and that the set of events is complete.[1]

The concept of probability is so simple and universal that it is hard to find a definition which is more than tautology. It is a function mapping the set of events onto the interval $(0, 1)$. Its simplest properties, often called the Kolmogorov axioms (Jaynes and Bretthorst, 2003, Appendix A) reflect the definition of events, i.e that the probability of the whole space is 1, and that the probability of the union of two disjoint events is the sum of their individual probabilities. If the events are part of a continuous space we can also define a probability density function (PDF) so that the probability that $x \in (a, b)$ is the integral

---

[1]this definition of event already takes us down one fork of the major distinction in statistics roughly described as Bayesians vs Frequentists. This is a Bayesian definition, a Frequentist will usually refer to an event as the outcome of an experiment.



or area under the PDF between $a$ and $b$. Formally we define

$$p(x) = \lim_{\delta x \to 0} \frac{P(\xi \in (x, x + \delta x))}{\delta x} \tag{1}$$

or its multi-dimensional counterpart.

## 2.2 Bayesian and non-bayesian inference

At this point any discussion of statistics bifurcates into two apparently incompatible methodologies, roughly termed Frequentist and Bayesian. Debate between the two schools has been carried on with surprising vigour. See Salsburg (2001) for a general history and Jaynes and Bretthorst (2003) for a partisan view. Characterizing the two schools is far beyond our scope here, especially as we will follow only one of them in much detail, but since even the language of the methods is almost disjoint it is worth introducing the new reader to the major concepts.

Frequentists generally pose problems as the quest for some property of a population. The property must be *estimated* from a sample of the population. *Estimators* are functions which compute these estimates from the sample. The design of estimators with desirable properties (no bias, minimum uncertainty etc) is a significant part of the technical apparatus of frequentist inference. Often we seek a physical parameter. This is not a bulk property but a single value which must be deduced from imperfect data. Here we treat each experiment as observing a member of a population so we can use the same apparatus; the

method is summed up in the phrase "the Replicate Earth Paradigm" (Bishop and Abramowitz, 2013).

  Bayesian statisticians concern themselves with the state of knowledge of a system. They regard the task of inference as improving this state of knowledge. We generate knowledge of some property of the system by applying some mathematical operation to the state of knowledge of the underlying system. Although we may call this estimating some property of the system the calculations involved are quite different from the estimators of frequentist statistics. As a practical example a frequentist

may estimate a mean by averaging his sample while a Bayesian may calculate an integral over her probability distribution. It is also important to note that for a Bayesian the state of knowledge does not necessarily come only from the sample itself. This use of information external or *prior* to the data is the most important practical difference between the two methods.

  Throughout this volume we will generally follow a Bayesian rather than frequentist approach. This represents the balance of activity in the field but is not a judgment on their relative merits.

# 3   Towards a Consistent Notation

The notation in use in this field is as varied as the nomenclature. This is unavoidable where similar techniques have been developed independently to answer the needs of many different fields. This diversity complicates the task for novices and experts alike as they compare literature from different fields. Producing a notation at once compact enough to be usable and general enough to cover all cases is a difficult task. Ide et al. (1997) made an attempt focused on numerical weather and ocean

prediction. Experience has shown their notation to be sufficient for most practical cases and we have followed it here. The



**Table 1.** Table of notation used throughout the issue. Borrows heavily from Ide et al. (1997) Appendix 1.

Basic Notation

| Symbol | Description |
| --- | --- |
| **BOLD** | Matrix |
| **bold** | Vector |
| $\mathbf{x}$ | Target variables for assimilation |
| $\mathbf{z}$ | Model state variables |
| $\mathbf{y}$ | Vector of observations |
| $J$ | Cost function |
| $\mathbf{U}(\mathbf{x}, \mathbf{x}^\mathrm{t})$ | Uncertainty covariance of $\mathbf{x}$ around some reference point $\mathbf{x}^\mathrm{t}$ |
| $\mathbf{C}(\mathbf{x})$ | Uncertainty correlation of $\mathbf{x}$ |
| $p(x)$ | Probability density function evaluated at $x$ |
| $G(\mathbf{x}, \mu, \mathbf{U})$ | Multivariate normal (Gaussian) distribution with mean $\mu$ and covariance $\mathbf{U}$ |
| $H$ | Observation operator mapping model state onto observables |
| $\mathbf{H}$ | Jacobian of $H$, often used in its place, especially for linear problems |
| $M$ | Model to evolve state vector from one timestep to the next |
| $\mathbf{M}$ | Jacobian of $M$ |
| $(\cdot)^\mathrm{a}$ | Posterior or analysis |
| $(\cdot)^\mathrm{b}$ | Background or prior |
| $(\cdot)^\mathrm{f}$ | Forecast |
| $(\cdot)^\mathrm{g}$ | (First) guess in iteration |
| $(\cdot)^\mathrm{t}$ | True |
| $\delta$ | Increment |

Some Useful Shortcuts

| Symbol | Description |
| --- | --- |
| $\mathbf{d}$ | $\mathbf{y} - H(\mathbf{x})$ (Innovation) |
| $\mathbf{U}(\mathbf{x})$ | Uncertainty covariance of $\mathbf{x}$ about its own mean, i.e. the true uncertainty covariance |
| $\mathbf{B}$ | $\mathbf{U}(\mathbf{x}^\mathrm{b})$ (Prior uncertainty covariance) |
| $\mathbf{Q}$ | $\mathbf{U}(\mathbf{x}^\mathrm{f}, \mathbf{x}^\mathrm{t})$ (Forecast uncertainty) |
| $\mathbf{R}$ | $\mathbf{U}_{(\mathbf{y} - H(\mathbf{x}^\mathrm{t}))}$ |
| $\mathbf{A}$ | $\mathbf{U}(\mathbf{x}^\mathrm{a})$ (posterior uncertainty covariance) |

notation does have an explicitly Bayesian flavour. For example we eschew the use of hats to describe estimates since we regard estimation as an operation on a probability distribution.

## 4 Fundamental Bayesian Theory





In this section we will sketch the fundamental solution of the inverse problem. This will motivate a more detailed description of the various inputs in the following section.

Tarantola (2004) states at the outset of his book that the state of knowledge of a system can be described by a probability distribution or corresponding probability density. Our task is to combine knowledge from measurements with pre-existing knowledge on the system's state to improve knowledge of the system. [2]

The recipe for combining these two sources of knowledge is traditionally introduced via conditional probabilities and expressed as Bayes' Theorem Laplace (1774). Tarantola (2004) and Jaynes and Bretthorst (2003) derive the machinery directly from the axioms of probability.

We postulate that the true state of the system is consistent with our prior knowledge of it and the measurements we take. The link between the system's state and the observations is created by an observation operator.[3]. Both observations and observation operators are imperfect and we describe the state of knowledge of them both with probability distributions.

The case for one system variable and one observation is described in Figure 1, slightly adapted from Rayner (2010). The left-hand panel shows the joint probability distribution for the system state (blue rectangle) and the measured quantity (red rectangle). Our imperfect knowledge of the mapping between them is shown by the green rectangle. In this simple case the three PDFs are uniform so there is no privileged value for the data or system state.

The solution of the problem is the intersection or multiplication of the three PDFs. Adding the requirement that the system state must take a value, we obtain the solution (right hand panel) by projecting the joint PDF onto the x-axis and normalizing it. We write the multiplication as

$$p(x) \propto p(x|x^{\mathrm{b}}) \times p(y^{\mathrm{t}}|y) \times p(y^{\mathrm{t}}|H(x)) \qquad (2)$$

The constant of proportionality follows from normalisation $\int p(x)\, dx = 1$. The last term in Equation 2 is the probability of a true value $y^{\mathrm{t}}$ given a value of the system state $x$ and is often termed the likelihood. We stress that the system state and the true value of the measured quantity are particular values. Our knowledge of them is instantiated in the PDFs for $x$ and $y^{\mathrm{t}}$. What we usually think of as the estimated state of the system $x^{\mathrm{a}}$ is some derived property of $p(x)$, e.g. its mode or maximum likelihood. Note that if any PDF is mischaracterized or if Equation 2 is not well applied, $x^{\mathrm{a}}$ can correspond to a very small probability, possibly even smaller than for $x^{\mathrm{b}}$. Such a situation may remain unnoticed in the absence of independent direct validation data for $x^{\mathrm{a}}$, as is currently the case for global atmospheric inversions. This issue underlines the necessity to keep a clear link with the statistical framework, despite the possible technical complexity of data assimilation systems. Note also that a by-product of the solution is a refined PDF for $\mathbf{y}^{\mathrm{t}}$, i.e. improved knowledge of the true value of the mesured quantity. The idea of a measurement being improved by a model is surprising at first.

## 5 The Ingredients

In this section we describe the various inputs to Equation 2 along with some closely related quantities.

---

[2]Tarantola (2004) uses the term information instead of knowledge. We avoid it here since it has a technical meaning.

[3]Rayner et al. (2010) refers to this as a model but here we reserve that term for the dynamical evolution of the system




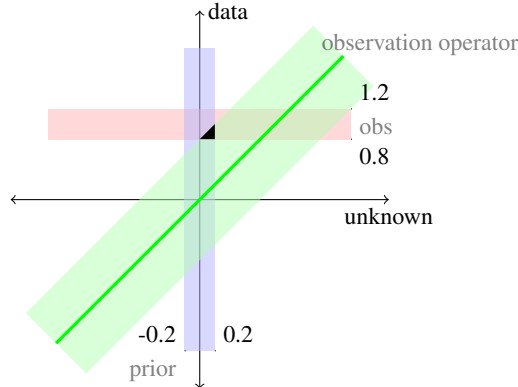

Joint probability distribution for system state (x-axis) and measurement (y-axis). The light-blue rectangle (which extends to infinity in the y direction) represents prior knowledge of the unknown system state. The light-red rectangle (which extends to infinity in the x direction) represents the knowledge of the true value of the measurement. The green rectangle (extending to infinity along the one-to-one line) represents the state of knowledge of the observation operator. The black triangle (the intersection of the three PDFs) represents the solution as the joint PDF.

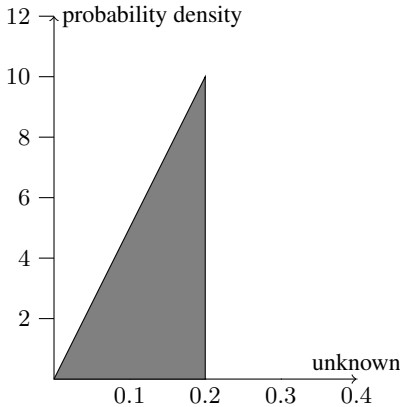

PDF for unknown system state obtained by projecting black triangle from the left panel onto the x-axis (i.e. integrating over all values of the observation).

**Figure 1.** Illustration of Bayesian inference for a system with one state variable and one observation. The top panel shows the joint probability distribution for the system state and the observation while the bottom panel shows the resulting PDF for the system state. Figure modified from Rayner (2010).



## 5.1 Deciding on Target Variables

Although not usually considered part of the problem it preconditions much of what follows. It is also the subject of some common misunderstandings and there is some definite guidance for the choice.

In general the target variables should include anything which is important to the behaviour of the system and is not perfectly known. The choice should not depend on us guessing which target variables can be constrained by the observations. There are three reasons for this:

1. We often wish to calculate the uncertainty in some predicted quantity. Limiting the number of target variables may underestimate this uncertainty;

2. We should let the data determine which variables can be constrained rather than guessing in advance (Wunsch and Minster, 1982);

3. If a target variable which should be important for explaining an observation is removed, the information from that observation may contaminate other target variables (Trampert and Snieder, 1996; Kaminski et al., 2001).

In an ideal world one would:

1. Decide on a target quantity of interest. This need not be a target variable for the assimilation; frequently it is a forecast from the model.

2. Use sensitivity analysis to find variables to which it was sensitive (e.g. Moore et al., 2012). This may often include forcings which are usually neglected.

3. Generate PDFs for the most sensitive variables and, with these, calculate the PDF in the quantity of interest generated by each potential variable. These two steps should never be separated, the sensitivity to a given variable is best expressed in uncertainty units for that variable.

4. Target the most sensitive variables with the cut-off being computationally driven

5. Calculate the PDF for the quantity of interest.

It is noteworthy how much calculation may be required before we start an assimilation.

## 5.2 Possible Prior Information

The PDF for the prior should reflect whatever information we have about the quantity before a measurement is made. This is not always easy since science is in continual dialogue between existing knowledge and new measurements and it is likely that measurements quite similar to the ones we are about to use have already informed the prior. We should also use any information we have, even if this is only obvious constraints such as positivity. A great deal of the reticence about Bayesian methods comes from the possible impact of the prior PDF on the posterior. This influence can be subtle. For example Jeffreys (1957) pointed



out that even using a positive scaling factor with a uniform distribution as a prior may have an influence on the posterior He recommended using a PDF which is uniform in the logarithm of the parameter.

In practice, prior information is most often taken from a model providing a mechanistic or process-based representation of the state (e.g. Gurney et al., 2002). The advantage of this approach is that it explicitly incorporates scientific understanding of the functioning of the state into the data assimilation process. Further, model outputs can easily be tailored to the inversion problems, with data being provided for a wide range of temporal or spatial resolutions. The disadvantage is that many alternate such representations often exist, and that the choice among them can have a major and scale-dependent influence on the final estimates (e.g. Lauvaux et al., 2012; Carouge et al., 2010), while the relative merits of such alternate priors is difficult to objectively and quantitatively determine when no ground truth is available. The availability of ground truth allows drawing the statistics of model performance, but not necessarily directly at the spatial and temporal scales of the prior information. In this case, one needs to study the upscaling or downscaling properties of these statistics, for instance through correlations (e.g. Chevallier et al., 2012).

Alternatives that reduce the reliance on potentially subjective prior information do exist, where less restrictive prior assumptions are made. The case of uniform priors described above is one such example. One practical example in biogeochemical data assimilation is the so-called "geostatistical" approach (e.g. Michalak et al., 2004), where the prior is described as a function of unknown hyperparameters that link ancillary datasets to the state, and where the prior uncertainty covariance is formally described through spatial and temporal correlation analysis that is based as strongly as possible on available observations. The advantage here is the decreased reliance on possibly subjective prior information (e.g. Gourdji et al., 2012; Miller et al., 2013), while the disadvantage is a more limited formal link to scientific process-based prior understanding of the system.

It is important to note that the various approaches used to construct the prior in data assimilation problems are not disjoint options, but rather can be thought of as points along a continuum of options that reflect different choices about the assumptions that a researcher is willing to make about the system (e.g. Fang et al., 2014; Fang and Michalak, 2015). The choice of prior is an expression of a choice of assumptions. With a careful framing of the specific question to be addressed, and with an explicit examination of the assumptions inherent to different possible approaches, one can design a set of priors that optimally blends existing approaches for a given application.

### 5.3 Observations

The PDF for the observation represents our knowledge of the true value of the measured quantity given the measurement. Note that neither the measurement nor the true value are random variables, it is only our state of knowledge that introduces uncertainty. The characterization of this PDF is the task of metrology and another paper in the volume will be dedicated to relevant aspects. An important point to make at this stage is that verification of a measurement means, for our purposes, verification of the PDF (including its mean, i.e. studying possible measurement systematic errors). That is it is even more important that the PDF we use *does*, in fact, represent the state of knowledge of the true value than that the value is as precise as possible.



### 5.4 Observation Operators

The PDF for the observation operator represents our state of knowledge of the true value of a simulated quantity that arises from a given state of the system. It includes any artifact caused by the resolution of the target variables Kaminski et al. (2001); Bocquet (2005). This PDF is often hard to verify since there are few cases where we know precisely the relevant values of the system state so that when we see a difference between the simulated and true value it is hard to assign this error to the observation operator or the prior. Frequently we must abstract the observation operator and run it under controlled conditions. An example is deliberate tracer release experiments (e.g. van Dop et al., 1998). Absent such direct verification calculations like sensitivity analyses or ensemble experiments (e.g. Law et al., 1996) give incomplete guidance. It is important to note that if the PDFs are Gaussian then the errors due to observations and observation operators may be added quadratically (Tarantola, 2004, Eq. 1.104). We frequently shorthand this as the data uncertainty (or worse data error) when it is usually dominated by the observation operator. The resulting PDF describes the difference we might expect between the simulated result of the observation operator and the measured value. Thus analysis of the residuals (observation − simulated quantity) can help test the assumed errors. This forms part of the diagnostics of data assimilation treated in Michalak and Chevallier (2016).

### 5.5 Dynamical Models

Although they are not one of the ingredients in figure 1, dynamical models are so common in this field that we include them here. Many of the systems studied are dynamical, that is they involve the evolution of the system state through time. We reserve the term dynamical model (often shorthanded to model) for the mathematical function or computer system that performs this evolution. We frequently consider the dynamical model to be perfect in which case the set of system states as a function of time are a unique function of the boundary and initial conditions. Some formulations relax this condition in which case we need a PDF for the model. In the most common case of a first-order model, the PDF represents our state of knowledge of the true state of the system at timestep $i + 1$ given its state at timestep $i$ and perfect knowledge of all relevant boundary conditions. As with observation operators this is hard to generate in practice since we rarely know the preexisting state or boundary conditions perfectly. In repeated assimilation cases like numerical weather prediction the statistics of many cases can give guidance. The PDF can also be deduced from the analysis of the model-data misfits when the statistics of the uncertainty of the model inputs and of the measurement errors are known Kuppel et al. (2013).

The boundary between the observation operator and the dynamical model depends on our choice of system variables. If we consider only the initial state and assimilate observations over a time-window (the classic case of 4-dimensional variational data assimilation) then the dynamical model forms part of the mapping between the unknowns and the observations so is properly considered part of the observation operator. If we include system state from the assimilation window (weak-constraint 4dvar) then the dynamical model enters as a separate object with its own PDF. We will see shortly how this can be incorporated explicitly into the statistical framework.

### 5.6 Hyperparameters



There is a generalization of the theory outlined in Section 4. Although the inputs to the inference algorithm are probability distributions, the forms of these distributions, and the observation operator and dynamical model are fixed. It can be useful or occasionally necessary to relax this constraint. If these PDFs are described by parameters then these parameters themselves can be assigned PDFs.

For example, if our prior PDF is a Gaussian of the form $p(x) = G(\mathbf{x}^{\mathrm{b}}, \mathbf{B})$ then we can write

$$p(x|\mathbf{B}) = G(\mathbf{x}^{\mathrm{b}}, \mathbf{B})P(\mathbf{B}) \qquad (3)$$

From such an expression we can infer properties of $p(\mathbf{B})$ such as the maximum-likelihood estimate (MLE). Approaches such as generalized cross-validation (GCV) which adjust the relative weight of prior and data contributions to a least squares cost function are equivalent since weighting a cost function is equivalent to changing an uncertainty. Krakauer et al. (2004) used

GCV to improve the consistency of an inversion while Michalak et al. (2005) solved for the MLE for various parameters of $\mathbf{R}$ and $\mathbf{B}$. Ganesan et al. (2014) solved sequentially for $p(\mathbf{B})$ in a so-called hierarchical approach in which $p(\mathbf{B})$ is used to generate $p(\mathbf{x})$. We note that adjusting $\mathbf{B}$ is a common approach to apparent inconsistency between the three input PDFs. The observation operator can also be absorbed into the generation of posterior PDFs, either by including model parameters in the target variables or introducing an index variable on an ensemble of models. Rayner (2016) applied this technique to an

ensemble of models used in an atmospheric inversion.

## 6   Solving the Inverse Problem

### 6.1   General Principles

We saw in Section 4 that the solution of the assimilation problem consisted of the multiplication of three PDFs to generate the posterior PDF. Before describing solution methods applicable in various circumstances we point out some general conse-

quences of Eq. 2. We begin by stressing that the posterior PDF generated in Eq. 2 is the most general form of the solution. Derived quantities are generated by operations on this PDF.

Second, we see that the only physical model involved is the forward observation operator. All the sophisticated machinery of assimilation is not fundamental to the problem although we need it to derive most of the summary statistics.

Third there is no requirement that a solution exists or, if it does, it can have vanishingly low probability. In Figure 1 this

would occur if various of the prior PDFs were made tighter (smaller rectangles for example) so that they did not overlap. This is a critical scientific discovery since it tells us our understanding of the problem is not consistent with the data, notwithstanding all the uncertainties of the inputs. It points out a boundary of "normal science" as defined by Kuhn (1962). [4] A later paper in this issue will focus on diagnosing such inconsistencies.

### 6.2   Sampling Methods

---

[4] We thank Ian Enting for pointing this out





A general approach for characterizing the posterior probability distribution of the target variables is to devise an approach for generating a large number of equally likely samples, or realizations, from this multivariate posterior distribution. We briefly describe a few example approaches that fall under this umbrella.

One approach to solving Eq. 2 is direct calculation of the product for a large number of realizations drawn from the constituent PDFs. Most simply, one can treat the PDF like any other scalar function and map it. Kaminski et al. (2002) did this for one dimension by tacitly assuming Gaussian PDFs for their models and data. This mapping approach requires exponential computations of the forward model, however, so is infeasible for even moderately dimensioned problems. In addition, most of these calculations will correspond to very low probability regions.

A range of more sophisticated Monte Carlo sampling approaches exist for multi-dimensional problems. The most straightforward of these is direct sampling of the posterior pdf, feasible for the case where this posterior pdf is sufficiently simple to allow sampling from the distribution. This would be the case, for example, for multivariate Gaussian or lognormal distributions.

In cases where direct sampling is not feasible, the strategy often becomes one of sequentially sampling from a surrogate distribution (often termed the proposal distribution) in such a way that the ensemble of samples ultimately represents a sample from the more complex target distribution. Arguably, the simplest of the approaches falling under this umbrella is rejection sampling. In this approach, a sample is generated from a proposal distribution that is simple enough to be sampled directly, and that sample is accepted or rejected with a probability equal to the ratio between the likelihood of the sampled value based on the target versus the proposal distribution, normalized by a constant (e.g. Robert and Casella, 2004).

A particularly useful class of Monte Carlo sampling strategies are Markov chain Monte Carlo (MCMC) approaches. In these approaches, the samples form a Markov chain, such that each sample is not obtained independently, but rather as a perturbation of the last previously accepted sample. More formally, a Markov Chain is a Stochastic traversal of some manifold in which the current state is dependent only on the immediately preceding state. The various MCMC approaches differ in how the candidate samples are generated based on the last accepted sample. Usually we construct the new state by adding a random perturbation to the previous state. For example, Metropolis et al. (1953) proposed and Hastings (1970) proved that a Markov chain based on the following algorithm produces a sample of an underlying PDF:

1. At iteration $i$ we have state $x_i$ with probability $p(x_i)$;

2. Generate a new candidate point $x_{i+1}^*$ by adding a random perturbation to $x_i$;

3. Choose a uniformly distributed random number $u \in [0,1]$;

4. If $p(x_{i+1}^*)/p(x_i) > u$ we accept $x_{i+1}^*$ as part of the solution $x_{i+1}$ and return to step 1, otherwise reject it and return to step 2.

The critical step is step 4 which is known as the Metropolis rule, and the algorithm overall is known as the Metropolis-Hastings algorithm. This algorithm produces samples with high serial correlation since successive points in the Markov Chain are likely to be close to each other in parameter space. The degree of correlation depends on the size and type of the perturbation applied in Step 2.



Rather than sampling the posterior pdf directly, one can also sample each of three PDFs sequentially (Figure 1). For this we can use the cascaded Metropolis algorithm (see Tarantola, 2004, Section 6.11). Once we have an accepted point from step 4 above we proceed to apply the Metropolis rule to the other probability distributions in the problem. In practice, if we are solving simultaneously for unknowns and data (the case in Figure 1) then we find a point acceptable to the model and data PDFs and only then go on to apply the model test. This is convenient since the model calculation is the most expensive part of the algorithm.

Another commonly-used MCMC algorithm, the Gibbs sampler, can be seen as a special case of the Metropolis Hastings algorithm where the target variables are sampled individually and sequentially, rather than sampling the full multi-variate pdf of target variables as a single step (e.g. Casella and George, 1992). An advantage of the Gibbs sampler is that it can lead to a higher rates of acceptance, because defining strategic approaches for generating perturbations is more straightforward for univariate distributions. A disadvantage is that the overall computational cost can still be higher due to the need to perform sampling multiple times for each overall realization.

The most important implementation detail for applying MCMC approaches is the choice of the perturbation (Step 2 in the case of the Metropolis Hastings algorithm) since this determines how the PDF is sampled. Good strategies are adaptive, lengthening the stepsize to escape from improbable regions of the parameter space and shortening it to sample probable parts (noting the comments on independence above).

Note that it takes at least one forward run for every accepted point meaning only inexpensive models can be treated this way. Furthermore the original algorithm cannot be parallelized since each calculation requires the previous one. On the other hand MCMH is exceedingly robust, can handle a wide variety of PDFs and requires no modification to the forward model.

### 6.3 Summary Statistics

We often describe the solution of the inverse problem using summary statistics of the posterior PDF. This can be because we assume that the posterior PDF has an analytic form with a few parameters which can describe it (e.g. the mean and standard deviation of the Gaussian) or because we need a numerical representation for later processing or comparison. This nearly universal practice is valuable but requires care. For example, if the target variables include a timeseries then we should not use the temporal standard deviation of the most likely estimate as a measure of the temporal variability of the timeseries. The uncertainty of the timeseries will add to the variability. The same is true of spatial variability. There are some cases which are so common they deserve explicit treatment however, provided the caveat above is borne in mind.

### 6.4 The Linear Gaussian Case

If $\mathbf{H}$ is linear and the PDFs of the prior, data and model are Gaussian then the posterior PDF consists of the multiplication of three Gaussians. We multiply exponentials by adding there exponents so it follows that the exponent of the posterior has a quadratic form and hence the posterior is Gaussian.



The Gaussian form for the inputs allows one important simplification without approximation. We noted in Section 4 that the solution to the inverse problem consisted of finding the joint posterior PDF for unknowns and observed quantities then projecting this into the subspace of the unknowns. As we saw in section 4 we also calculate the posterior PDF for $\mathbf{y}^{\mathrm{t}}$ the measured quantity. If there are very large numbers of observations this is computationally prohibitive and usually uninteresting.

If the PDFs are Gaussian (Tarantola, 2004, Section 6.21) showed that the last two terms of Eq. 2 can be combined into a single PDF $G(H(\mathbf{x}) - \mathbf{y}, \mathbf{R})$ where $\mathbf{R} = \mathbf{U}(H(\mathbf{x}) - \mathbf{y}^{\mathrm{t}}) + \mathbf{U}(\mathbf{y} - \mathbf{y}^{\mathrm{t}})$, i.e. by adding the two variances. The PDF now includes only the unknowns $\mathbf{x}$ and the posterior PDF can be written

$$p(\mathbf{x}) \propto G(\mathbf{x} - \mathbf{x}^{\mathrm{b}}, \mathbf{B}) \times G(\mathbf{H}(\mathbf{x}) - \mathbf{y}, \mathbf{R}) \qquad (4)$$

this generates a gaussian for $\mathbf{x}$ whose mean and variance can be calculated by a range of methods (Rodgers, 2000, ch.1).

The calculations of the mean and variance for the posterior PDF are separable i.e. we can calculate each without the other. Furthermore the posterior covariance $\mathbf{A}$ does not depend on $\mathbf{x}^{\mathrm{b}}$ or $\mathbf{y}$ so that one can calculate posterior uncertainties before measurements are taken (Hardt and Scherbaum, 1994). This is the basis of the quantitative network design approaches described in Kaminski and Rayner (2016).

The largest computation in this method is usually the calculation of $\mathbf{H}$ which includes the response of every observation

to every unknown. This may involve many runs of the forward model. Once completed $\mathbf{H}$ is the Green's function for the problem and instantiates the complete knowledge of the resolved dynamics. It enables a range of other statistical apparatus (e.g. Michalak et al., 2005) or Rayner (2016).

### 6.5   Dynamically Informed Priors: the Kalman Filter

A common case in physical systems has the state of the system evolving according to some dynamical model $M$. This is usually

(though not necessarily) some form of first-order differential equation. In such a case a natural algorithm for constraining the evolution of the system with data is

1.  At any step $n$ our knowledge of the system is contained in the probability distribution $p(\mathbf{x}^n)$;

2.  Calculate a probability distribution for the new state of the system $p(\mathbf{x}^{\mathrm{f},n})$ using the dynamical model $M$;

3.  Use $p(\mathbf{x}^{\mathrm{f},n})$ as the prior PDF for the assimilation step described in Section 4;

4.  This yields a posterior PDF $p(\mathbf{x}^{n+1})$ which we use as the input for the next iteration of step 1.

For the simplest case of linear $M$ and $H$ with Gaussian PDFs the algorithm was derived by Kalman (1960). The most difficult step is the generation of $p(\mathbf{x}^{\mathrm{f},n})$. For the original Gaussian case where $p(\mathbf{x}^n) = G(\mathbf{x}^n, \mathbf{B}^n)$ this is given by

$$p(\mathbf{x}^{\mathrm{f},n}) = G(M(\mathbf{x}^n), \mathbf{M}\mathbf{B}^n\mathbf{M}^{\mathrm{T}} + \mathbf{Q}) \qquad (5)$$

where $\mathbf{Q}$, the forecast error covariance, represents the uncertainty added to the state of the system by the model itself. The

matrix product in Equation 5 makes this approach computationally intractable for large numbers of unknowns and various



approaches have been tried using a well-chosen subspace. The ensemble methods discussed in Section 7.3 have generally supplanted these. This also holds for nonlinear variants of the system.

The original conception of the Kalman Filter was for dynamical control in which the state of the system is continually adjusted in accordance with arriving observations. For data assimilation our motivation is to hindcast the state of the system and, optionally, infer some underlying constants (e.g. Trudinger et al., 2007, 2008). A consequence of making inferences at each time separately is that the system may be forced to follow an incorrect observation. For a hindcast we can counter this by expanding our set of unknowns to include not only the current state but the state for several timesteps into the past. This technique is known as the Kalman Smoother (Jazwinski, 1970) and also exists in ensemble variants (Evensen, 2009) and in variational ones (see Section 7.1). Evensen (2009) also showed that the Kalman Filter described above is a special case of the Kalman Smoother with only one time-level considered.

## 7 Computational Aspects

Much of the computational machinery of data assimilation aims to estimate various summary statistics of the posterior PDF. The utility of this approach depends on whether these parameters summarize accurately the posterior PDF. It is best to choose the target summary statistic based on knowledge of the posterior PDF; there is no point estimating two parameters if the posterior distribution is described by only one (e.g. the Poisson Distribution). Here we limit ourselves to discussions of the two most common parameters the maximum likelihood estimate and the posterior covariance. Solving for these parameters does not itself assume a Gaussian posterior but we often do assume such normality.

### 7.1 Iterative Solution for the Mode: Variational Methods

The mode of the posterior PDF is the value of the unknowns which maximizes the posterior probability. If we can write

$$p(x) \propto e^{-J(x)} \tag{6}$$

then maximizing $p$ is equivalent to minimizing $J$.[5] In the linear Gaussian case $J$ is quadratic so minimizing it is termed a least squares solution. It is perhaps unfortunate that many treatments of data assimilation start from the discussion of a least squares problem and thus hide many of the assumptions needed to get there.

The minimization of $J$ takes us into the realm of numerical methods, beyond our scope. From the numerical viewpoint $J$ is a scalar function of the unknowns $\mathbf{x}$. Minimizing $J$ is a problem in the calculus of variations and so the methods are usually termed variational.

The quadratic form is sufficiently common to warrant more development. From Eq. 4, $J$ takes the form

$$J = \frac{1}{2} \left[ (\mathbf{x} - \mathbf{x}^{\mathrm{b}})^T \mathbf{B}^{-1} (\mathbf{x} - \mathbf{x}^{\mathrm{b}}) + (\mathrm{H}(\mathbf{x}) - \mathbf{y})^T \mathbf{R}^{-1} (\mathrm{H}(\mathbf{x}) - \mathbf{y}) \right] \tag{7}$$

Most efficient minimization algorithms require the gradient of $J$ ($\nabla_{\mathbf{x}} J$). In general these gradients are efficiently calculated by the use of adjoint methods (Griewank, 2000). For simpler cases we can use modified versions of $\mathbf{H}$ as follows: differentiation

---

[5]We cannot always write the PDFs this way, e.g. the uniform PDFs in Figure 1. In such cases the mode may not be defined.



of Eq. 7 yields

$$\nabla_{\mathbf{x}} J = \mathbf{B}^{-1}(\mathbf{x} - \mathbf{x}^{\mathrm{b}}) + \mathbf{H}^{\mathrm{T}}(\mathbf{R}^{-1}[H(\mathbf{x}) - \mathbf{y}]) \tag{8}$$

The first term in Eq. 7 derives from the prior PDF and will be zero if $\mathbf{x} = \mathbf{x}^{\mathrm{b}}$. The second term derives from the combined
PDF of model and observations. It is driven by $\mathbf{Hx} - \mathbf{y}$ (the mismatch between observations and simulations) weighted by
the inverse observational uncertainty covariance $\mathbf{R}^{-1}$. This is multiplied by $\mathbf{H}^{\mathrm{T}}$. $\mathbf{H}^{\mathrm{T}}$ is, put roughly, a reversed version of $\mathbf{H}$
mapping observation-like quantities back into the space of unknowns. For certain cases such as atmospheric transport such
models can be formulated explicitly (Uliasz, 1994; Seibert and Frank, 2004).

### 7.2 Calculation of Posterior Uncertainty

Any reasonable summary of a posterior PDF will involve an estimate of its spread as well as location. This spread almost always
has a higher dimension than the mode since it must describe the joint probabilities of parameters. This immediately raises a
problem of storage and computation since even moderate assimilation problems may include 100,000 unknowns for which
representation of the joint PDF is intractable. In purely Gaussian problems it can be shown that the Hessian or 2nd derivative
$\nabla_{\mathbf{x}}^2 J$ evaluated at the minimum is the inverse covariance of the posterior PDF. This is promising since many gradient-based
algorithms for minimization calculate low-rank approximations of the Hessian. Unfortunately they usually commence with the
largest eigen-values of the Hessian corresponding to the lowest eigen-values (best-constrained parts) of the covariance. When
convergence is achieved we are left with a subspace of the unknowns for which we can comment on the posterior uncertainty
and ignorance beyond this. Properly we should restrict analysis of the results to this limited subspace but it usually does not
map conveniently onto the physical problem we are studying (Chevallier et al., 2005).

An alternative approach is to generate an ensemble of modes using realizations of the prior and data PDFs (Fisher, 2003;
Chevallier et al., 2007). The resulting realizations of the posterior PDF can be used as a sample for estimating population
statistics. For Gaussian problems it can be shown that the sample covariance so generated is an unbiased estimator of the
posterior covariance.

### 7.3 The Ensemble Kalman Filter

The main limitations of the traditional Kalman Filter for data assimilation are its assumption of Gaussian PDFs and the difficulty
of projecting the state covariance matrix forward in time. The ensemble approach described in Section 7.2 can also be used to
circumvent these problems. The hierarchy of approaches is reviewed in Chen (2003). The most prominent (and most closely
related to the original) is the Ensemble Kalman Filter (NKF) (Evensen, 2003, 2009). Instead of representing exactly the PDF
by its mean and covariance we generate an ensemble of realisations. We replace Equation 5 by advancing each realisation with
the dynamical model then perturbing the result with noise generated from $\mathbf{Q}$. We then update each ensemble member using the
Gaussian theory described in Section 6.4. Equation 4 shows that, as well as a prior estimate (for which we use the ensemble
member), we need observations, an observational covariance and a prior covariance. The prior covariance we calculate from
the ensemble itself. The observational covariance we assume. It is important to note that we do not use the observations directly





but rather perturb these with the observational noise. This makes an interesting comparison with the simple linear Gaussian case where we use the observations directly. The explanation for the difference is that the observational noise contributes to the posterior uncertainty which, in the case of the NKF is calculated from the ensemble members.

Computationally we need to run a dynamical model for each realisation of the ensemble. This differs from Equation 5 where the propagation of the uncertainty scales with the number of unknowns. Another advantage is that, while Equation 5 is exact only for linear models, the ensemble method may capture nonlinear impacts on the state covariance. the biggest disadvantage is the sampling problem for the posterior PDF. The limited ensemble size introduces errors in the magnitudes of the variances and spurious correlations among state variables.

## 7.4 Combining Filtering and Sampling: the Particle Filter

Particle Filters or Sequential Monte Carlo methods relax the assumptions inherent in the NKF. Like the NKF, the Particle Filter samples the underlying PDFs with a series of realisations rather than propagating the mean and variance of the PDF. There are steps in the NKF which assume Gaussian probability distributions and the Particle Filter avoids these, seeking to fit the true underlying PDF. We parallel the description of the Kalman Filter algorithm to highlight the differences.

1. At any step $n$ our knowledge of the system is contained in the probability distribution $p(\mathbf{x}^n)$. Generate a set of realisations (usually called particles) $\mathbf{x}_i$ drawn from $p(\mathbf{x})^n$;

2. Propagate each realisation forward in time using the dynamical model $M$ to generate realisations of the forecast state $\mathbf{x}_i^m$;

3. Account for the uncertainty in the dynamical model by perturbing $\mathbf{x}_i^m$ according to the model PDF to generate realisations $\mathbf{x}_i^f$ consistent with the dynamical model and its uncertainty;

4. For each $\mathbf{x}_i^f$, generate the simulated observations $\mathbf{y}_i$ using the observation operator $H$ then evaluate the PDF for the observations $p(\mathbf{y})$ at the point $\mathbf{y}_i$, these probabilities act as weights for $\mathbf{x}_i^f$;

5. Using some numerical procedure with inputs of $\mathbf{x}_i^f$ and $p(\mathbf{y}_i)$ generate a posterior PDF $p(\mathbf{x}^{n+1})$ which we use as the prior PDF for the next iteration.

We see that the last three steps mirror the multiplication of PDFs described in Equation 2. Also the difference between the NKF and Particle Filter lies chiefly in how we generate the posterior PDF from the ensemble of forecast states; the NKF uses the linear Gaussian approximation while the Particle filter uses some kind of sampling. Thus the Particle Filter is a hybrid of the Kalman Filter and sampling approaches.

The relaxation of the Gaussian assumption is at once a strength and weakness of the Particle Filter. It can handle more complex PDFs than the NKF but it also requires many more samples since it samples the posterior PDF rather blindly while the NKF is guided by the Kalman update step into regions of high probability. This difference has meant that, while the NKF has been used successfully on some very large problems (such as numerical weather prediction) the Particle Filter has been





limited to the same problem sizes as the approaches described in Section 6.2. Recent developments such as the Marginal Particle Filter (Klaas et al., 2005) or the Equivalent Weights Particle Filter (Ades and van Leeuwen, 2015) try to overcome the sampling problem by introducing additional steps for the resampling of the particles.

An overview of the principal ideas of particle filtering and improvements upon the basic filter is given by Doucet et al.
(2001) while van Leeuwen (2009) reviews its application in geophysical systems. Stordal et al. (2011) points to some further generalisations which may combine advantages of the NKF and Particle Filter.

## 8   Conclusions

There are an apparently wide range of techniques available for assimilating data into models such as those used in biogeochemistry. All of these, however, inherit the basic formalism of Bayesian inference in which posterior probabilities are constructed
as conjunctions or multiplications of probability distributions describing prior knowledge, measured quantities and models relating the two. The methods differ in the information they assume or adopt on the various components. In general the more restrictive the assumptions the more powerful is the statistical apparatus available to analyze the system.

### Code and Data Availability

This paper contains some sketches of mathematical development but does not rely on any particular piece of code or data set.

*Acknowledgements.*  We acknowledge the support from the International Space Science Institute (ISSI). This publication is an outcome of the ISSI's Working Group on "Carbon Cycle Data Assimilation: How to consistently assimilate multiple data streams". Rayner is in receipt of an Australian Professorial Fellowship (DP1096309).



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
