# Peer review of "Fundamentals of Data Assimilation"

_Geoscientific Model Development, 2016_

## Short Comment (SC1) · 12 Jul 2016

This is a very useful article. Please find in the supplement a version of the manuscript with a few questions, a few comments, and a few small errors directly corrected.

Please also note the supplement to this comment:
http://www.geosci-model-dev-discuss.net/gmd-2016-148/gmd-2016-148-SC1-supplement.zip

---

## Referee Comment (RC1) · Anonymous Referee #1 · 22 Jul 2016

Comments on "Fundamentals of Data Assimilation" by Rayner et al

General Comments

The authors embark on an important mission, namely to provide a common language with which all data assimilation practitioners in biogeochemistry can speak with one another, as well as detailing certain techniques for this. Though this reviewer acknowledges the difficulty of the task, the present effort is a bit rough around the edges and needs quite a bit of work before it will be useful to the community at large. As such, I recommend major revision before publication.

Generally speaking, my major complaints about the manuscript are threefold: 1) There are very few places where examples of the techniques described being used in biogeochemistry are cited, except those by the authors themselves. Readers of this document would be best served to have a list of examples in hand to better understand

how the field has employed these techniques. Particularly in Section 7, where specific algorithms are detailed, at least a modest list of relevant papers is called for. 2) The manuscript seems to meander between vague generality and over-specificity. In particular, the bits that describe ideas relevant to the relaxation of prior assumptions and dependence on priors are much more specific and thoroughly cited than the rest. 3) The discussion of MCMC methods in 6.2 would be better placed with the material in section 7, which should be renamed something like "Implementations of the Theory" or something similar. This is a computational approach to sampling the posterior distribution, rather than a "general principle". This is likely to help the reader to better understand the notion of the posterior distribution as "the solution", but it will also likely tempt them to believe that this sampling method is the "right way" to get at the solution, which is certainly a problem specific conclusion to draw.

Specific Comments

Page 2, Lines 29-: The task of defining probability measures for the non-specialist is certainly nontrivial. The discussion of probability density function here is a bit confusing, especially since most will not understand what you mean by "continuous space", which I believe is that every interval in (0,1) has a preimage in the probability space, so that it makes sense to define a CDF. Without having to go into Radon-Nikodym derivatives, it's enough to define a CDF, and then define the PDF as its derivative, which is what you're doing. Perhaps a rearrangement of the words here would serve this purpose.

Table 1, entry for R: I believe the notation should be U(y-H(xt)), where the parenthetical bit is not a subscript.

Section 4: citation of Tarantola (2004) should be for the year 2005. I also think that given the heavy reliance on his developed theory, it may be worth pointing to his original 1982 paper as well as the edition of his book from 1987 (?), both of which are more readable by those new to the field.

[Figure]

Page 7, Line 16: Remove "target" from the first sentence and merely reference a quantity of interest.

Page 7, Line 18: It's probably worth stating outright that the PDF being computed is a marginal PDF, since you later say in step 5 to calculate the PDF for the quantity of interest. Another reason for this computation is that it's the product of the sensitivity and uncertainty that matter, and ensures an "apples to apples" comparison between different potential parameters.

Page 8, Line 2: Though the example is instructive, I'm not sure what purpose the last sentence serves, unless more information about the recommendation is given, such as what he's trying to optimize with this choice.

Page 8, Line 19: "more limited formal link" If the point is to remove reliance on subjective priors, then what are they being replaced with? A more honest sentence would be something like "replacing a formal link with an empirical one" or something similar.

Page 9, Line 7: "Absent such direct verification calculations like sensitivity analyses or ensemble experiments give incomplete guidance" This sentence would read better if the "like" were replaced with a comma.

Page 10 Equation 3: Should it be $p(B)$ rather than $P(B)$?

Page 10 Line 6: Should it be $p(x|B)$ rather than $p(B)$? I'm not sure how we infer the MLE of $p(B)$ from equation 3.

Page 13 Lines 14-17: This seems like a very good place to cite the synthesis inversion literature as a great body of examples of this technique for the biogeochemistry applications.

Page 15, Line 27 Across all fields, the common nickname for these techniques is "EnKF". To enable readers to connect this text to others in their area of specialty, it seems using the more common name would be most useful.

[Figure]

Page 15, Lines 32 to Page 16, Line 3: This was true for the initial formulations of the EnKF by Evens and others. Modern implementations favor a "deterministic" formulation that doesn't perturb observations, such as the Ensemble Adjustment Kalman Filter (EAKF) and the Ensemble Square Root Filter (EnSRF). Tippett et al (2004) is a good reference for this topic.

Page 16, Line 6: "ensemble method may capture nonlinear impacts on the state co-variance" I have heard this but never seen evidence. Is there a relevant citation? Mathematically, the covariance in equation 5 is exactly the covariance of forecasted state, using the jacobian rule for propagating uncertainty.

Page 16, Line 15: p(x)ˆn should be p(xˆn)

——————————————————

---

## Referee Comment (RC2) · Anonymous Referee #2 · 6 Aug 2016

**Review of 'Fundamentals of Data Assimilation,' by P. Rayner, A.M. Michalak and F. Chevallier**

**Summary of review**

The article aims to provide an outline of the fundamentals of data assimilation to biogeochemists. While the intention is good, I find that the authors have missed the opportunity of reconciling the approaches taken by different researchers in a 'review fashion', and it does not focus on biogeochemistry. Instead, the focus of the article is on the description of data assimilation from a statistical viewpoint. From this perspective, I find that the authors have some fundamental misconceptions, and that others have made presentations (see below) that are much clearer and achieve the necessary rigor. Further, while I appreciate the authors' wish to avoid a lot of mathematical notation, I find their technical presentation and the logic hard to follow on many occasions.

I hope the authors find the following comments helpful.

**General comments**

- The authors may not be aware of some articles that already discuss data assimilation from the same viewpoint. The most relevant is, to the best of my knowledge, that by Wikle and Berliner (2007), titled a 'Bayesian tutorial for data assimilation', which adopts the following working definition of data assimilation: 'An approach for fusing data (observations) with prior knowledge (e.g., mathematical representations of physical laws; model output) to obtain an estimate of the distribution of the true state of a process'. This working definition is the same as that of the authors of the present manuscript. Wikle and Berliner (2007) discuss Bayesian inference, the choice of prior distributions, Kalman filtering, particle filters, MCMC, Bayesian hierarchical models, and they give a lot of intuitive examples. What is the contribution of this article over that of Wikle and Berliner (2007)? Furthermore, there are other relevant works related to data assimilation from the meteorological sciences (e.g., Bocquet et al., 2010), that are not cited. Additionally, quantitative network design based on posterior uncertainties has been done elsewhere (e.g., Krause et al., 2008); see Pg.13 l.13.

- I found the omission of 'mathematically precise language' confusing, and despite the good intentions I am not convinced this is good practice. There are several problematic parts in the article. I just give a few examples:

  - Equation (2) is problematic. First, the left-hand side has to be a function of $y$ and $y^t$ unless one is presuming that these are known constants. Second, one usually seeks either the joint posterior distribution $p(y^t, x \mid y)$ or, depending on the scope, the posterior distributions $p(x \mid y)$ and $p(y^t \mid y)$. Both $y^t$ and $x$ are not directly observed. If (2) is referring to the posterior distribution $p(x \mid y)$, then one must take into account the fact that $y^t$ is unknown. This will inevitably involve an integral over

$y^t$ on the right-hand-side. Also, what role is $p(y^t \mid y)$ playing in (2) if it is not a function of $x$? (Such terms are usually absorbed into the constant of proportionality).

– Pg.5: The sentence '$x^a$ can correspond to a very small probability' cannot be interpreted when $x$ is endowed with a probability *density* function. The follow-up clause 'possibly even smaller than for $x^b$' needs to be qualified – under what distribution are you comparing probabilities?

– Pg.8: The sentence 'Note that neither the measurement nor the true value are random variables, it is only our state of knowledge that introduces uncertainty' comes across as a hybrid Bayesian/frequentist argument. I agree that the distributions may be constructed based on state of knowledge (this is the subjective argument for eliciting distributions), but in the Bayesian framework both $x$ and $y$ are random variables. One may condition on an observed $y$ to carry out inference on $x$, but this is different from saying that $y$ is not a random variable.

– Equation (3) and its subsequent description are incorrect. First, I suspect that the authors wanted the left-hand-side to be a posterior (or conditional) distribution. Second, the right-hand side is the factorized joint distribution. Third, one cannot obtain an MLE from (3), only posterior inferences (except in the special case that the prior distribution is uniform).

- As outlined in the previous point, equations (2) and (3) are incorrect; further, Figure 1 is very hard to interpret (see below). I believe these misconceptions arise because the authors have not placed data assimilation into a hierarchical modelling framework as Wikle and Berliner (2007) did (although the manuscript mentions the hierarchical model once on Pg.10). The authors need to condition on a set of data for inference on the state, and I was not able to find where they do this (see also (4)). One can also view data assimilation as a state-space estimation problem which is another connection that is not made.

**Specific comments**

- Pg.1 l.14 and Pg.7 l.9: The authors talk about a model 'choice', but in a Bayesian setting care is needed, and uncertainty arising from the consideration of multiple models has to be taken into account.

- Pg.1 l.17: I agree that 'data assimilation', 'parameter estimation', 'inverse modelling', and 'model-data fusion', are often used interchangeably, but I thought this article should not ignore this source of confusion, rather it should take the opportunity to resolve it.

- Pg.1 l.20: It should be made clear that the model's predictive performance should be assessed on out-of-sample data and not just any data.

- Pg.2 l.30: What are $x$ and $\xi$?

- Pg.3 l.13: The likelihood function, which is important to both frequentists and Bayesians, needs to be considered in such a discussion.

- Pg.6: Figure 1 (top) is misleading. The axes have arrows in both directions (what does this mean?), words are used for axes labels (I assume the 'unknown' is $x$, the data $y$, but then where is $y^t$?). This figure aims to instruct but it is very difficult to relate to the mathematics.

- Pg.7 l.18: Does 'Generate PDFs' mean 'Elicit prior PDFs'? Does 'Calculate the PDF for the quantity of interest' mean 'compute the posterior PDF?'. Since the authors are advocating a Bayesian approach they need to be more precise in their terminology.

- Pg.7 l.29: Jeffreys (1957) is given as a reference but it is not in the reference list.

- Pg.8 l.2: This is incorrect. The objective Jeffrey's prior is highly dependent on the parameter being inferred and the model.

- Pg.8 l.11: It is not clear what the sentence 'upscaling or downscaling properties of these statistics, for instance through correlations' is implying.

- Pg.9 l.2: The use of 'simulated quantity' in this context is misleading – I believe the authors mean $H(x)$ as the 'simulation' but it could also mean 'simulation of a random quantity'.

- Pg.9 l.9: The statement on adding the errors quadratically is both mathematically and statistically incorrect. First, one must be operating on a log scale, and second, this statement should be considering covariances and not errors.

- Pg.12: The discussion on MCMC is misleading. First, the sentence 'An advantage of the Gibbs sampler is that it can lead to a higher rate of acceptance' is inaccurate. The Gibbs sampler ensures that the acceptance rate is exactly 1 (guaranteed acceptance). Second, increased computational cost of the Gibbs sampler is not only due to the required multiple sampling, but also due to high intra-chain correlation. Finally, MCMC is *not* exceedingly robust. In fact it is quite a messy approximate inference approach, as applied Bayesians will attest to.

- Pg.13 l.10: It is incorrect that one can calculate the posterior mean without knowledge of the posterior covariance.

- Pg.13 l.14: Given the previous discussion it should have been mentioned that in a Bayesian framework one may (and should) also invoke prior distributions on the forward models, since this is usually a highly uncertain component.

- Pg.14 l.16: It is not a maximum likelihood estimate but a maximum-a-posteriori estimate. This difference is crucial in this context.

- Pg.15 l.27: It should be 'EnKF' not 'NKF'.

**Concluding remark**

All in all, after accounting for the works of Tarantola (2005) and Wikle and Berliner (2007) and for the authors' misconceptions, I do not see the added value of this article. It would be much more valuable to the community if it were transformed into a 'review article' of methods used in biogeochemistry (illustrating how those methods fit into a common biogeochemical framework). The title would need to be changed to reflect this and the contents would need to reflect the considerable work already done in Bayesian connections to data assimilation.

**References**

Bocquet, M., Pires, C. A., and Wu, L. (2010). Beyond Gaussian statistical modeling in geophysical data assimilation. *Monthly Weather Review*, 138:2997–3023.

Krause, A., Singh, A., and Guestrin, C. (2008). Near-optimal sensor placements in Gaussian processes: Theory, efficient algorithms and empirical studies. *Journal of Machine Learning Research*, 9:235–284.

Tarantola, A. (2005). *Inverse Problem Theory and Methods for Model Parameter Estimation*. SIAM, Philadelphia, PA.

Wikle, C. K. and Berliner, L. M. (2007). A Bayesian tutorial for data assimilation. *Physica D: Nonlinear Phenomena*, 230:1–16.

---

## Referee Comment (RC3) · Anonymous Referee #3 · 14 Sep 2016

The authors of this manuscript are experts in applying data assimilation methods to problems in the geosciences, especially in the area of interpreting trace gas measurements in terms of surface sources and sinks. In this manuscript they discuss several popular data assimilation methods – variational data assimilation, Kalman filters, and particle filters – in the context of Bayes theorem, showing how, at root, they are all essentially solving the Bayesian inference problem in different ways. They do this using a common notation, derived from the Ide et al (2003) paper, and discuss differences in the assumptions of each approach. An example of one such difference that is whether dynamical errors are considered for the transport model or not.

As will be seen below in my detailed comments, I had a problem with the notation used in this manuscript, especially with the use of "yˆt", or the "true measurement" to represent a measurement vector. I would prefer a notation in which the terms "truth" and "true" refer to quatities that have specific, objective real values, as opposed to

error distributions associated with them. In data assimilation and estimation theory in general, it is usually the estimates of these "true" quantities, which have errors and thus error distributions associated with them. Here, instead, we have a "true" measurement with an error distribution associated with it – a confusing concept to one familiar with the standard estimation theory notation and concepts. Perhaps this notation is coming direct from the Ide reference – if so, this manuscript would have been a good place to correct it and set forth a notation that attributes the error more correctly to the quantities the error is more appropriately placed on. More generally, this seems to be related to the dropping of the notation indicating which quantities are estimates versus those that are not. I don't think this distinction can be attributed to the difference between the Frequentist and Bayesian viewpoints – what is given here just seems wrong to me. The notation problems lead to problems in the way the underlying issues are thought about, in my opinion.

A second issue is the discussion of dynamical errors. The authors do discuss the issue in Section 5.5, but for much of the manuscript they stick with a notation that rolls together the transport model and the observation operator into a single function H(x). This conflation of the two error sources into a single term was established early in the atmospheric inversion literature, unfortunately, and has often not been un-conflated even today. In other fields, however, the two were never conflated, and a general discussion such as this should do its best to keep them separated – frequently, this has not been the case in this manuscript, as is reflected below in my detailed comments.

In general, this manuscript is a helpful discussion of the inter-relatedness of several different estimation methods used in the geosciences and would be worth publishing. I hope the authors will consider my comments as providing a path for improving the manuscript further. There are some notational issues that must be corrected before publication (e.g. the use of x vs. z for the state).

Detailed comments:

p2 L29: Shouldn't the probability of the union of two disjoint events be the product of the probability of each, not the sum?

p5 L12: You seem to be using "system variable" interchangeably with "state variable"? Might it not be clearer just to stick with "state variable"?

p5 L19: Equation (2) and the discussion in the text. Here, you seem to be using variable 'x' to represent the state of the system, though in your notation in Table 1, you say you are using 'z' for that. It seems that you need to say that your target variable 'x' is in fact the same as the state of the system 'z', so that you can reasonably write 'H(x)' instead of the 'H(z)' that you put forth in the _text_ (but not the notation) in Table 1 (i.e. "H Observation operator mapping model state onto observables"). Also, since we are on this topic, is "target variable" meant to be synonymous with "control variable" (that is, the vector of those parts of the system that can be controlled or manipulated to get the desired outcome), used often in the control theory literature? If so, it would have been better for Ide (and you here?) to reserve 'z' for that, and use 'x' for the state variable (consistent with that existing control theory literature). What's done is done, I suppose...

p5 L21: "the system state x" – again, inconsistent with what you have in Table 1, where the state is "z"

p5 L28: "a refined PDF for ŷt": This talk of a PDF for ŷt I think is mis-conceived. You state yourself that "We stress that the system state and the true value of the measured quantity are particular values. Our knowledge of them is instantiated in the PDFs for x and yt." This makes it clear that there is a distinction between the true value itself (which doesn't have a distribution, but rather a fixed, actual value) and our best estimate of the true value, which is in error and has an error distribution. The blurring of the line between the two, which you have built into your notation here, is particularly unfortunate when it comes to the measurement, which you call ŷt. One can imagine that the system has a true state ẑt which, when measured perfectly (without error) would yield

the measurement value corresponding to that true state. It would be a natural extention of the concept of a "truth" to refer to this quantity as the "true" measurement; i.e. $\hat{y}t$ = $H(\hat{z}t)$, for the case where H is assumed to be a perfect measurement operator. In the real world, the measurement process is not perfect, the actual measurement would have errors (reflecting errors in our knowledge of how to make a proper measurement) that would cause these real measurements to deviate from the ideal measurement, the perfect measurement, the "true" measurement. If the errors in this measurement process were gaussian, one could specify a gaussian measurement uncertainty on each flawed measurement y, and use that to quantify the error between these real measurements and the "true" measurement that would be obtained in the absense of measurement error. Instead, the authors choose to use the notation $\hat{y}t$ for the flawed, real-world measurement, rather than the perfect measurement $H(\hat{x}t)$ (obtained with a perfect H). This may not make much of a difference if we are always dealing with the difference between an actual, flawed measurement and the underlying value that it is attempting to measure, $H(\hat{z}t)$, but from a conceptual standpoint, it is placing the label "true" on the wrong quantity and seriously confusing the issue. Those formulations that keep estimates separate from the underlying objective reality place the distinction between "truth" and error-affected estimates correctly with their notation, I think; the notation used here, in contrast, confuses where the error should be placed.

I would be much happier if the authors made a distinction between the "true" underlying measurement $H(\hat{z}t)$ (where H is perfect), and an actual measurement of that quantity, possibly affected by random measurement errors: that quantity is usually called something else, "z" for example, to indicate that it is a measurement prone to all the errors an actual measurement might have. Don't put the label "true" on that.

p5 L29: "The idea of a measurement being improved by a model is surprising at first." This can still be the case, but it would reflect an improvement of an _estimate_ y of the measurement rather than the true measurement $\hat{y}t$. When thought of in those terms, it is not surprising at all. Why it appears surprising here is that the authors have used

the notation "yˆt" for the measurement – with that notation, it does appear surprising that you can improve upon something that is already "true".

p7, L19-20, Since there are two variables being discussed, it is not clear which variable the uncertainties should be couched in terms of, in the last sentence.

p7 L25: It is not clear here whether "the quantity" that the covariance is being calculated for is the "quantity of interest" in item #1 of the second list above, or of a target variable. Following Table 1, it seems like we need to calculate the uncertainties in the target variables, x. Why are we interested in the PDF of some other variable, even if it is "of interest", if it does not factor into the estimation problem? My understanding of the assimilation problem, using the notation laid out in Table 1, is that the uncertainties tracked in the method are those for the targeted variables x. Those seem distinct from "the quantity of interest" discussed here. Why case the uncertainties back onto a variable that is not the target variable?

p8 L28-29: "Note that neither the measurement nor the true value are random variables, it is only our state of knowledge that introduces uncertainty." It is not clear what this means. One could think of the true measurement as having a single value, reflecting objective reality, and the measurement being a random variable, reflecting the uncertainty contained in the measurement/modeling process. Why could the measurement itself not be considered a random variable, in that case?

p9 L5: "difference between the simulated and true value" of the measurements: This may get at the root of the problem I was having above with the definition of yˆt. It seems that the notation "yˆt" is being used as the actual measurement, including any measurement noise or biases, rather than as that measurement that would be given by the measurement operator operating on the true state in the absense of any measurement noise or errors in the operator. I would suggest that this notation be changed to something else.

p9 L10-13: "We frequently shorthand this as the data uncertainty (or worse data error)

when it is usually dominated by the observation operator. The resulting PDF describes the difference we might expect between the simulated result of the observation operator and the measured value. Thus analysis of the residuals (observation − simulated quantity) can help test the assumed errors. This forms part of the diagnostics of data assimilation treated in Michalak and Chevallier (2016)."

I would agree with this statement if the observation operator includes only the error in going between the propagated state vector and the observation. If, however, it includes also the error in the propagated state vector (and thus error in the dynamical model), then it is confusing two sources of error that are best kept separate (as in the formulation of the Kalman filter). Confusion on this point is prevelent in our field, resulting in model-data mismatch uncertainties being inflated much more than is truly justified. I see that the authors go briefly into this issue below, but perhaps greater emphasis on this point would be justified.

p9 L20: You need a PDF for the model error, not the model.

p9 L26-31: You have shown here how dynamical errors may be considered in the context of one implementation (variational data assimilation). It might be worth mentioning another common implementation, sequential filters (like the Kalman filter): since the state is estimated repeatedly across short spans, the dynamical errors can be accounted for explicitly by inflating the estimate of the state covariance as the state is propagated forward by the model (this is in fact built into the standard Kalman filter development).

p9 L9: "are equivalent" – equivalent to what? Adding some commas in this sentence might help to make it clearer.

p10 L13: "The observation operator can also be absorbed into the generation of posterior PDFs". It is not clear on the surface what this means. Could you please be more specific/clear, so the reader does not have to consult the reference to understand what is being discussed?

p10 L22-23: "Second, we see that the only physical model involved is the forward observation operator. All the sophisticated machinery of assimilation is not fundamental to the problem although we need it to derive most of the summary statistics."

For time-dependent problems in which a dynamical model is used, this dynamical model would be a second physical model that should be involved (this is the case for most of our geostatistical applications). The fact that it often is not involved in the equations we write down is an error in the way we approach the problem (i.e. using a strong dynamical constraint instead of a weak one (in the variational approach) or using a Kalman filter with dynamical errors added explicitly). The lumping of dynamical errors together with errors in the observational operator is a gross approximation that results in conceptual errors of the sort made here in this statement.

p13 L3-4: "As we saw in section 4 we also calculate the posterior PDF for yt the measured quantity." An oblique reference to this was given at the very end of Section 4, but no calculation was given. Perhaps you should give an equation at the end of Section 4 showing how the posterior PDF for yt is calculated, to support this statement you make here.

p13 L13-14: "The largest computation in this method is usually the calculation of H which includes the response of every observation to every unknown. This may involve many runs of the forward model."

Again, here, you are addressing the specific case where you conflate the dynamical model and the observation operator into a single function H. This is a specific approximation to the general case, which keeps the two separate.

"Once completed H is the Green's function for the problem and instantiates the complete knowledge of the resolved dynamics." This is only the case if you are ignoring dynamical errors. As the authors themselves note earlier in the paper, a more sophisticated analysis would allow errors in these Greens functions due to dynamical errors. It is hard to accept that these Greens functions include "complete knowledge" if they

do not account for these dynamical errors.

p14 L3-4: "...in which the state of the system is continually adjusted...". Since the true state of the underlying system had a single trajectory through time (i.e. has some objective real value, rather than a probability distribution), what you are really referring to here is some estimate of the state of the system. I believe your language and notation should reflect that.

p14 L6-8: "For a hindcast we can counter this by expanding our set of unknowns to include not only the current state but the state for several timesteps into the past. This technique is known as the Kalman Smoother (Jazwinski, 1970)"

To be more specific, this technique of including a number of previous times in the state is referred to as a FIXED LAG Kalman filter. This should be mentioned, as there are at least a couple other flavors of Kalman smoothers (fixed point and fixed interval).

p14 L22-23: "It is perhaps unfortunate that many treatments of data assimilation start from the discussion of a least squares problem and thus hide many of the assumptions needed to get there." You are making an assumption yourself here – that those who are using the least squares method want to make detailed assumptions about the statistics for their problem. If all they care about is getting an unbiased estimate and minimizing the standard deviation of their errors (measurement and prior), irrespective of what the higher moments of the PDF might look like, then the least squares approach is consistent and works just fine.

p14 L25-26: "Minimizing J is a problem in the calculus of variations and so the methods are usually termed variational." What you say here is overly-specific and doesn't really capture the essence of the problem. Really, minimizing J is a minimization problem. There are many numerical methods for minimizing a cost functional, and most of them are not variational. One does not need to get into the calculus of variations to understand that if one goes down-gradient on a manifold, one will get closer to the minimum. Most of the standard minimization methods use this concept as their basis. True, the

calculus of variations allows one to calculate gradients in a computationally efficient manner, and those gradients can be used in gradient-based descent methods to do the minimization, but this does not make these descent methods "variational".

p15 L5-7: For the case we are using here, in which transport and measurement are conflated in H, HˆT is the adjoint.

p16 L6-8: Amonng the disadvantages of the EnKF, you might note that inflation is often added to the ensemble to prevent the spread of the ensemble members from collapsing. This is often done in an ad hoc manner. Thus, effectively, the dynamical noise that is added with physical meaning in the straight Kalman filter is replaced with an ad hoc inflation term that has lost its physical meaning.

p16 L13: please add "given in Section 6.5" after "We parallel the description of the Kalman Filter algorithm" to help the reader remember where this was

Corrections to grammar, punctuation, etc.:

p1 L8 : add a comma after 'debate'

p1 L9: the semi-colon should be a colon, I think

p1 L13: add a comma after "For example"

p2 L23: capitalise "P" in "Pg"

p2 L32: replace the comma with a semi-colon after "definition"

p3 L18: add a comma before "the calculation"

Table 1, line describing 'd': Since H acts on the state z, not the vector of target variables, this should read "y - H(z)"

Table 1, line describing 'R': First, the variables inside U() should not be subscripted, as they are now. Second, the quantity inside U() should be "y - H(zˆt)", for the same reason as above.

Table 1, line describing 'U(x)': I am familiar with this expressed as being the uncertainty of an _estimate_ of x around the true value of x, xˆt. Similarly for the definition of "x" up top, there is usually a distinction made between an estimate of a vector of target variables, and a simple listing of what those variables happen to be. You attribute this to the Frequentist view of the world and drop the distinction, but I think it is getting you in trouble here – perhaps you can get around this by mentioning some of this in the description of "U(x)" and what you are assuming in defining it this way. In other words, how do you answer if someone asks you what the difference is between a vector of target variables x and their true value, xˆt? Wouldn't the vector x be the vector of true values? If so, how can U(x) be defined, if the difference is always zero? If the vector of x is not the vector of true values of x (xˆt), then what is it a vector of? (If not of estimates, then of what?)

p5 L7: correct the Laplace citation (put all within parentheses?)

p5 L28: correct to "measured"

p6, L2: add a comma after "problem"

p7 L28: Another word besides "reticence" might be more appropriate. "reticence" means a hesitence to speak. It sounds like you want something reflecting a hesitance to use the Bayesian approach, or to trust it.

p8 L5: here I would use "system" rather than "state", since the state represents the underlying system, and it is the functioning of the system that we care about.

p8, last line: for clarity, I would suggest adding commas after the initial "That is" and after "true value". Adding a "rather" before "than that" would also help.

p9 L3-4: put the two references inside of parentheses.

p9 L4-6, sentence starting with "The PDF": some commas in here would help this read better.

p9 L7-8: "Absent such direct verification calculations like sensitivity analyses or ensemble experiments (e.g. Law et al., 1996) give incomplete guidance." The subject of this sentence appears to be missing. Please reword to clarify this.

p9 L18: add a comma after "perfect"

p9 L19: add a comma after "condition"

p9 L25: put the reference in ()

p10 L2: add a comma after "model" or remove the one after "distributions"

p10 L10: add a comma before "while"

p12 L18: add a comma before "meaning"

p12 L31: change "there" to "their" (or possibly "three"?)

p14 Eq (7): The capital H's should be italic here – no need to linearize yet.

p15 L27; p16 L3; p16 L10: "NKF" – do you mean "EnKF" here?

p16 L6: Capitalize to get "The biggest"

---

## Author Comment (AC1) · 9 Jan 2017

article times natbib

**Response to Referees' Comments**

Peter Rayner, Anna M. Michalak and Frédéric Chevallier

January 9, 2017

We thank the three anonymous referees and Thomas Kaminski for their comments which have allowed us to clarify various points in the paper. There are some common points made by several referees. We will deal with these first as general comments then with particular comments from each referee. there are also some comments which are about the paper rather than requesting specific changes and we will deal with these first. We place referees' comments in typewriter font and our replies in Roman

**General Comments**

`the work has been better done elsewhere` Referee II referred particularly to *Wikle and Berliner* (2007). This is indeed a fine paper and we thank the referee for pointing it out. It does, in our view, a rather different job to our manuscript. (*Wikle and Berliner*, 2007) is more concerned with methods of solution of the inverse problem, including some very helpful examples, while our interest is more on the ingredients of the problem and very broad classes of solution methods. Secondly the language of *Wikle and Berliner* (2007) is more mathematically demanding than our present manuscript. This is a strength since it allows a terse and unambiguous development. It is also likely

to be unreadable to much of our target audience. This topic is difficult and important and we would argue strongly there is room for a multiplicity of approaches to explaining it. We believe that our balance of accessibility at the cost of ambiguity is valuable as is that of *Wikle and Berliner* (2007) not to mention other papers like *Evans and Stark* (2002). We do commend the focus on hierarchical approaches in *Wikle and Berliner* (2007) and have strengthened our discussion of this area in the revised manuscript. We do not believe that recasting the problem in this light (the recommendations about Figure 1 and related text) yet meets our test of completeness versus accessibility.

The paper should be recast as a review of uses of DA in biogeochemistry. This is related to the comment that The literature is handled badly with too much focus on the authors' own work. We group these comments since our response to them is similar. We note that this paper forms part of a special issue. Special issues in Copernicus journals maintain a balance of connectivity and autonomy; papers must make sense standalone but the whole needs to be greater than the sum of the parts. The special issue concerns the state-of-the-art in biogeochemical DA, especially the problem of assimilating disparate data streams. Each paper will of necessity review key examples in its area. It would be duplicative to repeat that process here. Our task is to provide the necessary connections among the papers so that a reader across the issue has a framework to compare and contrast the papers. We discussed providing an annotated bibliography like that of *Rayner et al.* (2010) but ruled it out for the abovementioned reasons. That said, we have deepened and broadened the bibliography throughout the paper, with a series of key examples cited for each aspect we discuss. As a result the bibliography has roughly doubled in length.

The authors have some fundamental misconceptions about the underlying theory and this is represented in poor notation. Again we treat these together since they overlap. First we thank Reviewer II for pointing out the error in Equation 2. We had written the need for the integral in

the caption to Figure 1 and neglected it in Equation 2. We have corrected this and parallelled the development in Figure 1 by writing out the multiplication of probabilities then the integral in separate equations. Secondly there is disagreement about the nature of the true value of the observed quantity $y^t$. Physical causality motivates the statement that it is a single value but draws us into the epistemological minefield of truth existing independent of knowledge. We will therefore stop trying to make this distinction in Section 5 and simply refer to $y^t$ as a random variable describing our knowledge of the observed quantity.

For notation we are in something of a dilemma. Every writer in this field could come up with some variations in any existing notation which would improve it in their eyes. All too often they do. The result is a bewildering multiplicity. We made the choice that the imperfect notation of *Ide et al.* (1997) was a better choice than creating another imperfect notation of our own.

**Specific Comments**

Reviewer I

The manuscript seems to meander between vague generality and
over-specificity.  In particular, the bits that describe ideas
relevant to the relaxation of prior assumptions and dependence
on priors are much more specific and thoroughly cited than the
rest. We presume here the author refers to Section 5.6. We think the solution to this, and to a problem raised by Reviewer II is a brief section on hierarchical approaches since the hyperparameters we describe are a special case of a hierarchical description of the system. We have thus rewritten Section 5.6.

The discussion of MCMC methods in 6.2 would be better placed

with the material in section 7, which should be renamed
something like "Implementations of the Theory" or something
similar. We tried the reviewer's suggestion in an earlier draft. We didn't find it to
work as well but we have tried a different split of section 6.2 where the generalities
of sampling methods are introduced and a new subsection for Section 7 where the
computational approaches are outlined.

The task of defining probability measures for the
non-specialist is certainly nontrivial.  The discussion
of probability density function here is a bit confusing,
especially since most will not understand what you mean by
"continuous space", which I believe is that every interval in
(0,1) has a preimage in the probability space, so that it makes
sense to define a CDF. Without having to go into Radon-Nikodym
derivatives, it's enough to define a CDF, and then define the
PDF as its derivative, which is what you're doing.  Perhaps a
rearrangement of the words here would serve this purpose.    We
have rearranged this. We don't think it's a good idea to introduce any ideas from
measure theory to an article like this, beautiful as they are. We also also don't
necessarily want to define a CDF since not all our variables are ordered in a natural
way for such a definition.

Table 1, entry for R: I believe the notation should be
U(y-H(xt)), where the parenthetical bit is not a subscript.
Corrected.

Section 4:  citation of Tarantola (2004) should be for the
year 2005.  I also think that given the heavy reliance on his
developed theory, it may be worth pointing to his original 1982
paper as well as the edition of his book from 1987 (?), both of
which are more readable by those new to the field. Done. We have

also taken the chance to include some other notable texts.

`Remove "target" from the first sentence and merely reference a quantity of interest.` Done.

`It's probably worth stating outright that the PDF being computed is a marginal PDF, since you later say in step 5 to calculate the PDF for the quantity of interest. Another reason for this computation is that it's the product of the sensitivity and uncertainty that matter, and ensures an "apples to apples" comparison between different potential parameters.` We are still to perform the data assimilation so, except for correlations among different parameters, the joint PDF is the product of marginals so it's not clear what this adds. We have added "prior" instead. We have added the second comment.

`Page 8, Line 2: Though the example is instructive, I'm not sure what purpose the last sentence serves, unless more information about the recommendation is given, such as what he's trying to optimize with this choice.` Agreed, we now note the problem but not the solution.

`Page 8, Line 19: "more limited formal link" If the point is to remove reliance on subjective priors, then what are they being replaced with? A more honest sentence would be something like "replacing a formal link with an empirical on" or something similar.` We have added a phrase on replacing subjective with empirical information to reflect this comment.

`Page 9, Line 7: "Absent such direct verification calculations like sensitivity analyses or ensemble experiments give incomplete guidance" This sentence would read better if the "like" were replaced with a comma.` We have added a comma after

"verification" to clarify.

Page 10 Equation 3: Should it be p(B) rather than P(B)? Page 10 Line 6: Should it be p(x|B) rather than p(B)? I'm not sure how we infer the MLE of p(B) from equation 3. Yes, the reviewer is correct but we have rewritten this section to emphasize the hierarchical approach.

Page 13 Lines 14–17: This seems like a very good place to cite the synthesis inversion literature as a great body of examples of this technique for the biogeochemistry applications. We have added a series of references and a brief history of the use of these matrix methods.

Page 15, Line 27 Across all fields, the common nickname for these techniques is "EnKF". To enable readers to connect this text to others in their area of specialty, it seems using the more common name would be most useful. Agreed and changed throughout.

Page 15, Lines 32 to Page 16, Line 3: This was true for the initial formulations of the EnKF by Evens and others. Modern implementations favor a "deterministic" formulation that doesn't perturb observations, such as the Ensemble Adjustment Kalman Filter (EAKF) and the Ensemble Square Root Filter (EnSRF). Tippett et al (2004) is a good reference for this topic. We thank the reviewer for this and have added a reference to *Tippett et al.* (2003) and a note to this effect.

Page 16, Line 6: "ensemble method may capture nonlinear impacts on the state covariance" I have heard this but never seen evidence. Is there a relevant citation? Mathematically, the covariance in equation 5 is exactly the covariance of forecasted state, using the jacobian rule for propagating uncertainty. The problem is the time at which the model is evaluated. If we

linearise at $\mathbf{X}^n$ and propagate the covariance with the resulting Jacobian we will get a different result than if we propagated the ensemble forward. If we truly calculated the sensitivity of the state at time $n+1$ to the state at time $n$ with a complete tangent linear calculation and the nonlinear dynamics the equivalence noted by the reviewer would hold. Adding this complexity would probably confuse readers at the level we're targeting so we have not added this explanation to the text.

```
Page 16, Line 15:
```
$p(x)^n$ should be $p(x^n)$ corrected.

Reviewer II

Here we cover points not mentioned in the general response above.

```
- Pg.5:  The sentence "x
```
$x^{\mathsf{a}}$
```
can correspond to a very small
probability" cannot be interpreted when
```
$x$
```
is endowed with a
probability density function.  The follow-up clause "possibly
even smaller than for
```
$x^{\mathsf{b}}$
```
" needs to be qualified – under what
distribution are you comparing probabilities?
```
We have softened the language here. It is sufficient for our purposes to say that poor consistency between the PDF for the true and modelled value of the observation should prompt concern. We then come back to the point during the new section 5.6.

```
I believe these misconceptions arise because the authors have
not placed data assimilation into a hierarchical modelling
framework as Wikle and Berliner (2007) did (although the
manuscript mentions the hierarchical model once on Pg.10).  The
authors need to condition on a set of data for inference on the
state, and I was not able to find where they do this (see also
(4)).  One can also view data assimilation as a state-space
estimation problem which is another connection that is not
```

made. Maybe we have a simple problem of language here. Data should be thought of as the number returned by the measuring instrument. Under this definition data are not random variables except insofar as digital measurements have finite resolution hence return intervals. The true value of the observed quantity can be treated as a random variable provided one remembers one is discussing our knowledge of it. Conditioning this true value on the measured value is reasonable but adds little. The true value is conditioned on other things as well, concentrations must be positive for example. The hierarchical approach most certainly has value but we believe the place to introduce it is after the basic concepts have been explained rather than adding the extra complexity here. State estimation is covered where it most commonly occurs in biogeochemical applications, the Kalman Filter and its descendants.

Pg.1 l.14 and Pg.7 l.9: The authors talk about a model 'choice', but in a Bayesian setting care is needed, and uncertainty arising from the consideration of multiple models has to be taken into account. "has to be" is a stretch, there are very few papers in biogeochemical data assimilation in which the researcher has access to multiple models. Should it be considered? Undoubtedly. It is now discussed in the rewritten Section 5.6 and treated at length in another article in the issue.

Pg.1 l.17: I agree that 'data assimilation', 'parameter estimation', 'inverse modelling', and 'model-data fusion', are often used interchangeably, but I thought this article should not ignore this source of confusion, rather it should take the opportunity to resolve it. Resolving it is beyond our power since it requires control of how others use language. Pointing out that readers should not be confused by different terms for the same thing is the best we can do.

Pg.1 l.20: It should be made clear that the model's predictive performance should be assessed on out-of- sample data and not just any data. Note that we did not limit ourselves to "predictive performance".

There is clearly diagnostic information from the fit to the data used in the algorithm. The validation data is important enough though that we have added a comment to the sentence.

`Pg.2 l.30:  What are x and ξ?` $x$ is a point and $\xi$ an integration variable in Eq. (1)

`Pg.3 l.13:  The likelihood function, which is important to both frequentists and Bayesians, needs to be considered in such a discussion.` It is obviously an important concept but this does not seem the right place to introduce it, it is part of the apparatus while here we are discussing the aims.

`Pg.6:  Figure 1 (top) is misleading.  The axes have arrows in both directions (what does this mean?), used for axes labels (I assume the 'unknown' is x, the data y, but then where is $\mathbf{y^t}$)` We agree, using the same symbols as in the equations is clearer than using words, corrected.

`Pg.7 l.18:  Does 'Generate PDFs' mean 'Elicit prior PDFs'? Does 'Calculate the PDF for the quantity of interest' mean 'compute the posterior PDF?'.  Since the authors are advocating a Bayesian approach they need to be more precise in their terminology.` No in both cases.  At least as usually used elicitation is the process of assimilating expert knowledge. This is not how things usually proceed in biogeochemical DA where the prior information is as likely to be outputs from previous experiments or, even more common, from a process model. Similarly, the pdfs are those for the quantity of interest, not the posterior pdfs for the target variables. We have attempted to clarify this by adding "prior" and "posterior" to the relevant steps. The problem with the precision sought by the reviewer is it does not cover the range of cases that occur in the applications we consider.

**Pg.7 l.29: Jeffreys (1957) is given as a reference but it is not in the reference list.**

Corrected, it should have been Jeffreys 1939.

Pg.8 l.2: This is incorrect. The objective Jeffrey's prior is highly dependent on the parameter being inferred and the model. We have deleted the sentence from "he recommended"

Pg.8 l.11: It is not clear what the sentence 'upscaling or downscaling properties of these statistics, for instance through correlations' is implying. biosphere models can be run at different resolutions. We can calculate the uncertainties at coarser resolution by aggregation of the statistics at finer resolution. We have clarified this.

Pg.9 l.2: The use of 'simulated quantity' in this context is misleading – I believe the authors mean H(x) as the 'simulation' but it could also mean 'simulation of a random quantity'. We added the word "model" to clarify.

Pg.9 l.9: The statement on adding the errors quadratically is both mathematically and statistically incorrect. First, one must be operating on a log scale, and second, this statement should be considering covariances and not errors. We checked the proof in Tarantola Section 6.21 and think it is correct. For the second point the reviewer must have missed the adverb "quadratically" in the sentence. We have strengthened the point by replacing it with "by adding their covariances". We had cited the incorrect equation number though so we thank the reviewer for the note.

Pg.12: The discussion on MCMC is misleading. First, the sentence 'An advantage of the Gibbs sampler is that it can lead to a higher rate of acceptance' is inaccurate. The Gibbs sampler ensures that the acceptance rate is exactly 1 (guaranteed acceptance). Second, increased computational cost of the Gibbs sampler is not only due to the required multiple

sampling, but also due to high intra-chain correlation. The re-viewer is correct that in most applications of the Gibbs sample, the one-dimensional pdf for a single element of the state space is sampled directly, leading to an acceptance rate of 1. We have modified the passage to reflect this. The interchain correlation problem exists for Markov chain sampling methods in general so we have added a sentence about this in the description of sampling methods.

Finally, MCMC is not exceedingly robust. In fact it is quite a messy approximate inference approach, as applied Bayesians will attest to. The reviewer is stating opinion as fact here. We did, however, mean something more specific about dependence on distributional assumptions and have modified the text to be more explicit.

Pg.13 l.10: It is incorrect that one can calculate the posterior mean without knowledge of the posterior covariance. the reviewer is incorrect here. One of the standard forms for the posterior mean is

$$\mathbf{x} = \mathbf{x}^b + \mathbf{P} \times \mathbf{H}^\mathsf{T} \times (\mathbf{HPH}\mathsf{T} + \mathbf{R})^{-1} \times (\mathbf{y} - \mathbf{Hx}^b)$$

which nowhere includes a calculation of the posterior covariance. However the next sentence contains the important information that one does not need measurements or prior means to calculate the posterior covariance. Thus we deleted the disputed sentence. We also added the reference cited earlier by the reviewer.

Pg.13 l.14: Given the previous discussion it should have been mentioned that in a Bayesian framework one may (and should) also invoke prior distributions on the forward models, since this is usually a highly uncertain component. This point is now covered more extensively in the rewritten Section 5.6.

Pg.14 l.16: It is not a maximum likelihood estimate but a maximum-a-posteriori estimate. This difference is crucial in this context. Wording corrected.

Pg.15 l.27: It should be 'EnKF' not 'NKF'. Also noted by Reviewer I. Corrected throughout.

All in all, after accounting for the works of Tarantola (2005) and Wikle and Berliner (2007) and for the authors' misconceptions, I do not see the added value of this article. It would be much more valuable to the community if it were transformed into a 'review article' of methods used in biogeochemistry (illustrating how those methods fit into a common biogeochemical framework). The title would need to be changed to reflect this and the contents would need to reflect the considerable work already done in Bayesian connections to data assimilation. We again thank the reviewer for the careful review. Our considerable combined experience of teaching these methods has informed the balance of rigour and accessibility we have chosen and we hold to that choice. For example teaching the hierarchical approach as an extension rather than a foundation is a pedagogical not conceptual choice.

**0.1  Reviewer III**

We note again the divergence of views among the reviewers on notation and its implications for the underlying philosophy. We believe the problem is best resolved by the idea that there is a single true value of the unknowns just as there is for the measured quantities but that our knowledge of either is imperfect and is best expressed through PDFs. We have been careful to avoid the terms "estimate" or "estimator" in their technical sense since these terms mean something quite specific in Frequentist development and we don't wish to imply that meaning.

A second issue is the discussion of dynamical errors. The authors do discuss the issue in Section 5.5, but for much of

the manuscript they stick with a notation that rolls together
the transport model and the observation operator into a single
function H(x). This conflation of the two error sources into a
single term was established early in the atmospheric inversion
literature, unfortunately, and has often not been un-conflated
even today. In other fields, however, the two were never
conflated, and a general discussion such as this should do its
best to keep them separated – frequently, this has not been the
case in this manuscript, as is reflected below in my detailed
comments. We deal with this question explicitly in Section 5.5. In conventional
atmospheric inversions the transport model *is* the observational operator. There is
no dynamical model since there is no equation describing the time evolution of the
unknowns. Thus errors in atmospheric dynamics which one might normally consider
as dynamical errors are, here, rightly considered errors in the observational operator.
It is possible to include atmospheric state variables within the unknowns (e.g. *Zhang
et al.*, 2015) but this is still not common in this field and the EnKF method used in that
paper follows the formalism we describe in Section 7.3.

p2 L29: Shouldn't the probability of the union of two disjoint
events be the product of the probability of each, not the sum?
this is most easily understood by considering events as sets of points in the space
of the unknowns. then it becomes clear that the union of the two events is the union
of the two sets and the rest follows. This may be a question of terminology. We
follow *Tarantola* (2005) in using the term conjunction to describe two events occurring
together, probably the case the reviewer has in mind.

p5 L12: You seem to be using "system variable" interchangeably
with "state variable"? Might it not be clearer just to stick
with "state variable"? We were indeed being inconsistent but not in the
way the reviewer thought. We were using "target variable" and "system variable"

interchangeably. "State variable" means something which defines the state of the dynamical model. It may or may not be a target for the assimilation.

```
p5 L19:  Equation (2) and the discussion in the text.  Here,
you seem to be using variable 'x' to represent the state of
the system, though in your notation in Table 1, you say you
are using 'z' for that.  It seems that you need to say that
your target variable 'x' is in fact the same as the state
of the system 'z', so that you can reasonably write 'H(x)'
instead of the 'H(z)' that you put forth in the text (but
not the notation) in Table 1 (i.e.  "H Observation operator
mapping model state onto observables").  Also, since we are on
this topic, is "target variable" meant to be synonymous with
"control variable" (that is, the vector of those parts of the
system that can be controlled or manipulated to get the desired
outcome), used often in the control theory literature?  If so,
it would have been better for Ide (and you here?)  to reserve
'z' for that, and use 'x' for the state variable (consistent
with that existing control theory literature).  What's done is
done, I suppose.
```
We share the reviewer's regret about the plethora of notations in use. We did not help here by using the ambiguous "system variable" which we have replaced throughout. For the choice of variable names, all we can do is try to be unambiguous. The reviewer pointed out places where we fell short which we have corrected.

```
p5 L21:  "the system state x" - again, inconsistent with what
you have in Table 1, where the state is "z"
```
See previous point.

```
p5 L28:  "a refined PDF for yËEt":  This talk of a PDF for
yËEt I think is mis-conceived.  You state yourself that "We
stress that the system state and the true value of the measured
```

quantity are particular values.  Our knowledge of them is
instantiated in the PDFs for x and yt." This makes it clear
that there is a distinction between the true value itself
(which doesn't have a distribution, but rather a fixed, actual
value) and our best estimate of the true value, which is in
error and has an error distribution.  The blurring of the line
between the two, which you have built into your notation here,
is particularly unfortunate.  See general discussion throughout this re-
sponse. We believe this apparent problem evaporates if one regards the PDFs as
describing the state of our knowledge. Quantities like the distribution of the difference
between our best estimate of a quantity and its true value are not needed if we follow
this reasoning.

(continuing) when it comes to the measurement, which you call
yËEɟt.  One can imagine that the system has a true state zËEɟt
which, when measured perfectly (without error) would yield
the measurement value corresponding to that true state.  It
would be a natural extention of the concept of a "truth"
to refer to this quantity as the "true" measurement; i.e.
$\mathbf{y}^\mathbf{t}$  =   $H(\mathbf{z}^\mathbf{t})$, for the case where H is assumed to be a perfect
measurement operator.  In the real world, the measurement
process is not perfect, the actual measurement would have
errors (reflecting errors in our knowledge of how to make a
proper measurement) that would cause these real measurements to
deviate from the ideal measurement, the perfect measurement,
the "true" measurement.  If the errors in this measurement
process were gaussian, one could specify a gaussian measurement
uncertainty on each flawed measurement y, and use that to
quantify the error between these real measurements and the
"true" measurement that would be obtained in the absense of

measurement error. Instead, the authors choose to use the notation $\mathbf{y}^t$ for the flawed, real-world measurement, rather than the perfect measurement $\mathbf{H}(\mathbf{x}^t)$ (obtained with a perfect H). There are a couple of problems here. Firstly $\mathbf{H}$ is not a measurement, it is rather a simulation of the same quantity which is measured. More important is the reviewer's misunderstanding of $\mathbf{y}^t$. It is, as the reviewer hopes, the true value of the measured quantity. It has an associated PDF which represents our knowledge of it. As mentioned earlier the "flawed measurement" does not have a PDF since it represents a value returned by a measuring instrument.

(continuing) This may not make much of a difference if we are always dealing with the difference between an actual, flawed measurement and the underlying value that it is attempting to measure, H(zËEt), but from a conceptual standpoint, it is placing the label "true" on the wrong quantity and seriously confusing the issue. Those formulations that keep estimates separate from the underlying objective reality place the distinction between "truth" and error-affected estimates correctly with their notation, I think; the notation used here, in contrast, confuses where the error should be placed. I would be much happier if the authors made a distinction between the "true" underlying measurement H(zËEt) (where H is perfect), and an actual measurement of that quantity, possibly affected by random measurement errors: that quantity is usually called something else, "z" for example, to indicate that it is a measurement prone to all the errors an actual measurement might have. Don't put the label "true" on that. We think this point has been discussed in previous responses. It is clear that the formulation of the problem as states of information about the true values of measurements and target variables is posing problems. This paper is not the place for a treatise on the distinctions between

[Figure]

Bayesian and Frequentist interpretations of probability but we have added a warning that the formulation we borrow may surprise some readers.

```
p5 L29:  "The idea of a measurement being improved by a
model is surprising at first." This can still be the case,
but it would reflect an improvement of an estimate y of the
measurement rather than the true measurement yËEt.  When
thought of in those terms, it is not surprising at all.  Why
it appears surprising here is that the authors have used the
notation "yËEt" for the measurement – with that notation, it
does appear surprising that you can improve upon something that
is already "true".
```
If we replace the word "estimate" with the phrase "knowledge of" in the comment we would agree with this. We have expanded the text slightly to emphasise this.

```
p7, L19-20, Since there are two variables being discussed, it
is not clear which variable the uncertainties should be couched
in terms of, in the last sentence.
```
We have been more explicit about which variables we mean.

```
p7 L25:  It is not clear here whether "the quantity" that
the covariance is being calculated for is the "quantity
of interest" in item #1 of the second list above, or of a
target variable.  Following Table 1, it seems like we need to
calculate the uncertainties in the target variables, x.  Why
are we interested in the PDF of some other variable, even if
it is "of interest", if it does not factor into the estimation
problem?  My understanding of the assimilation problem, using
the notation laid out in Table 1, is that the uncertainties
tracked in the method are those for the targeted variables x.
Those seem distinct from "the quantity of interest" discussed
```

here. Why case the uncertainties back onto a variable that is not the target variable? Many users of scientific results can benefit from knowing the uncertainty of quantities such as climate predictions. If these are the results of an assimilation (the "quantity of interest") then we need a method for calculating their uncertainty. We describe this here. We have expanded the description a little to make this clearer.

p8 L28-29: "Note that neither the measurement nor the true value are random variables, it is only our state of knowledge that introduces uncertainty." It is not clear what this means. One could think of the true measurement as having a single value, reflecting objective reality, and the measurement being a random variable, reflecting the uncertainty contained in the measurement/modeling process. Why could the measurement itself not be considered a random variable, in that case? See previous comments on PDFs reflecting states of knowledge.

p9 L5: "difference between the simulated and true value" of the measurements: This may get at the root of the problem I was having above with the definition of yËEt. It seems that the notation "$\mathbf{y}^t$" is being used as the actual measurement, including any measurement noise or biases, rather than as that measurement that would be given by the measurement operator operating on the true state in the absense of any measurement noise or errors in the operator. I would suggest that this notation be changed to something else. We argue it is important that it *not* be changed so that readers can familiarise themselves with the underlying logic.

p9 L10-13: "We frequently shorthand this as the data uncertainty (or worse data error) when it is usually dominated by the observation operator. The resulting PDF describes the

difference we might expect between the simulated result of the
observation operator and the measured value.  Thus analysis
of the residuals (observation − simulated quantity) can help
test the assumed errors.  This forms part of the diagnostics of
data assimilation treated in Michalak and Chevallier (2016)."
I would agree with this statement if the observation operator
includes only the error in going between the propagated state
vector and the observation.  If, however, it includes also the
error in the propagated state vector (and thus error in the
dynamical model), then it is confusing two sources of error
that are best kept separate (as in the formulation of the
Kalman filter).  Confusion on this point is prevelent in our
field, resulting in model-data mismatch uncertainties being
inflated much more than is truly justified.  I see that the
authors go briefly into this issue below, but perhaps greater
emphasis on this point would be justified. This point is discussed in
Section 5.5. We have added text pointing out the relationship between the choice of
target variables and the ascription of errors.

p9 L20:  You need a PDF for the model error, not the model. No,
the original language is correct.  A PDF for the model error would indicate we are
uncertain what the model error should be. That is the subject of the next section.

p9 L26-31:  You have shown here how dynamical errors may be
considered in the context of one implementation (variational
data assimilation).  It might be worth mentioning another
common implementation, sequential filters (like the Kalman
filter):  since the state is estimated repeatedly across short
spans, the dynamical errors can be accounted for explicitly by
inflating the estimate of the state covariance as the state is

propagated forward by the model (this is in fact built into the standard Kalman filter development). **This is addressed in the relevant section, we have added a sentence pointing to it.**

p9 L9: "are equivalent" – equivalent to what? Adding some commas in this sentence might help to make it clearer. **We have rewritten this section so the sentence no longer exists.**

p10 L13: "The observation operator can also be absorbed into the generation of posterior PDFs". It is not clear on the surface what this means. Could you please be more specific/clear, so the reader does not have to consult the reference to understand what is being discussed? **We have rewritten this section.**

p10 L22-23: "Second, we see that the only physical model involved is the forward observation operator. All the sophisticated machinery of assimilation is not fundamental to the problem although we need it to derive most of the summary statistics." For time-dependent problems in which a dynamical model is used, this dynamical model would be a second physical model that should be involved (this is the case for most of our geostatistical applications). The fact that it often is not involved in the equations we write down is an error in the way we approach the problem (i.e. using a strong dynamical constraint instead of a weak one (in the variational approach) or using a Kalman filter with dynamical errors added explicitly). The lumping of dynamical errors together with errors in the observational operator is a gross approximation that results in conceptual errors of the sort made here in this statement. **The reviewer's concern is legitimate but beside the point here. The**

important point here is that we need only forward models to calculate the relevant probabilities or likelihoods. The question of what those forward models should be and the implications for what target variables should be included was dealt with in an earlier comment.

p13 L3-4: "As we saw in section 4 we also calculate the posterior PDF for yt the measured quantity." An oblique reference to this was given at the very end of Section 4, but no calculation was given. Perhaps you should give an equation at the end of Section 4 showing how the posterior PDF for yt is calculated, to support this statement you make here. this is a good idea, we have added relevant text in Section 4.

p13 L13-14: "The largest computation in this method is usually the calculation of H which includes the response of every observation to every unknown. This may involve many runs of the forward model." Again, here, you are addressing the specific case where you conflate the dynamical model and the observation operator into a single function H. This is a specific approximation to the general case, which keeps the two separate. We believe that the calculation of $\mathbf{H}$, the Jacobian of observations with respect to target variables will work whether or not the target variables include state variables from the observation operator. The reviewer's recurrent concern about the choice of model is best dealt with elsewhere.

p14 L3-4: "...in which the state of the system is continually adjusted...". Since the true state of the underlying system had a single trajectory through time (i.e. has some objective real value, rather than a probability distribution), what you are really referring to here is some estimate of the state of the system. I believe your language and notation should

reflect that. This comment probably refers back to previous concerns over notation. We have added the words "knowledge of" to clarify.

p14 L6-8: "For a hindcast we can counter this by expanding our set of unknowns to include not only the current state but the state for several timesteps into the past. This technique is known as the Kalman Smoother (Jazwinski, 1970)" To be more specific, this technique of including a number of previous times in the state is referred to as a FIXED LAG Kalman filter. This should be mentioned, as there are at least a couple other flavors of Kalman smoothers (fixed point and fixed interval). We have added "fixed lag".

p14 L22-23: "It is perhaps unfortunate that many treatments of data assimilation start from the discussion of a least squares problem and thus hide many of the assumptions needed to get there." You are making an assumption yourself here – that those who are using the least squares method want to make detailed assumptions about the statistics for their problem. If all they care about is getting an unbiased estimate and minimizing the standard deviation of their errors (measurement and prior), irrespective of what the higher moments of the PDF might look like, then the least squares approach is consistent and works just fine. The reviewer has misunderstood our point. We did not imply that people should not use least squares. We hold that starting a presentation of the methods with the minimisation of a least squares cost function hides too much of the statistical underpinnings.

p14 L25-26: "Minimizing J is a problem in the calculus of variations and so the methods are usually termed variational." What you say here is overly-specific and doesn't really

capture the essence of the problem.  Really, minimizing J
is a minimization problem.  There are many numerical methods
for minimizing a cost functional, and most of them are not
variational.  One does not need to get into the calculus of
variations to understand that if one goes down-gradient on
a manifold, one will get closer to the minimum.  Most of the
standard minimization methods use this concept as their basis.
True, the calculus of variations allows one to calculate
gradients in a computationally efficient manner, and those
gradients can be used in gradient-based descent methods to do
the minimization, but this does not make these descent methods
"variational".  This is a good point.  We have softened the language to
"commonly".

p15 L5-7:  For the case we are using here, in which transport
and measurement are conflated in H, HËET is the adjoint.  By this
stage in the development we haven't specified a particular problem such as atmo-
spheric inversion. That's first done in the next sentence and the undue specificity isn't
helpful here.

p16 L6-8:  Amonng the disadvantages of the EnKF, you might note
that inflation is often added to the ensemble to prevent the
spread of the ensemble members from collapsing.  This is often
done in an ad hoc manner.  Thus, effectively, the dynamical
noise that is added with physical meaning in the straight
Kalman filter is replaced with an ad hoc inflation term that
has lost its physical meaning. A good point, we have added text both on
inflation and localisation as ad hoc responses to incomplete posterior covariances.

p16 L13:  please add "given in Section 6.5" after "We parallel
the description of the Kalman Filter algorithm" to help the

reader remember where this was Done.

p1 L8 :   add a comma after 'debate' while "while" is a conjunction here which do not generally attract preceding commas in British English. We will leave this one for the copy editor.

p1 L9:   the semi-colon should be a colon, I think Not a colon but perhaps a comma.

p1 L13:   add a comma after "For example" done.

p2 L23:   capitalise "P" in "Pg" Done.

p2 L32:   replace the comma with a semi-colon after "definition" done.

p3 L18:   add a comma before "the calculation" done.

Table 1, line describing 'd':   Since H acts on the state z, not the vector of target variables, this should read "y - H(z)" Agreed, done.

Table 1, line describing 'R': First, the variables inside U() should not be subscripted, as they are now.   Second, the quantity inside U() should be "y - H(zËEt)", for the same reason as above. Agreed, done.

Table 1, line describing 'U(x)':   I am familiar with this expressed as being the uncertainty of an estimate of x around the true value of x, $\mathbf{x}^t$.   Similarly for the definition of "x" up top, there is usually a distinction made between an estimate of a vector of target variables, and a simple listing of what those variables happen to be.   You attribute this to the Frequentist view of the world and drop the distinction,

but I think it is getting you in trouble here – perhaps you can
get around this by mentioning some of this in the description
of "U(x)" and what you are assuming in defining it this way.
In other words, how do you answer if someone asks you what
the difference is between a vector of target variables x and
their true value, xËEₜt?  Wouldn't the vector x be the vector of
true values?  If so, how can U(x) be defined, if the difference
is always zero?  If the vector of x is not the vector of true
values of x (xËEₜt), then what is it a vector of?  (If not of
estimates, then of what?) **x**, as we define earlier in the table, really is a point
in the manifold of target variables. **x** is mainly used for defining functions on that
manifold. Superscripts refer to particular points in that manifold, the background, the
analysis, the true etc. estimates and estimators, if we use them at all, will enter the
problem very late as we describe elsewhere.

p5 L7:  correct the Laplace citation (put all within
parentheses?) done.

p5 L28:  correct to "measured" done.

p6, L2:  add a comma after "problem" done.

p7 L28:  Another word besides "reticence" might be more
appropriate.  "reticence" means a hesitence to speak.  It
sounds like you want something reflecting a hesitance to use
the Bayesian approach, or to trust it. Changed to reluctance.

p8 L5:  here I would use "system" rather than "state", since
the state represents the underlying system, and it is the
functioning of the system that we care about.  Changed, also ear-
lier in the same sentence.

p8, last line:  for clarity, I would suggest adding commas
after the initial "That is" and after "true value".  Adding
a "rather" before "than that" would also help. Agree about the com-
mas (and have added them) but not about "rather", "It is more important that x than y"
seems perfectly standard.

p9 L3-4:  put the two references inside of parentheses. done.

p9 L4-6, sentence starting with "The PDF":  some commas in here
would help this read better. Splitting the sentence seems a better choice.

p9 L7-8:  "Absent such direct verification calculations like
sensitivity analyses or ensemble experiments (e.g.  Law et al.,
1996) give incomplete guidance." The subject of this sentence
appears to be missing.  Please reword to clarify this.   "calcula-
tions" is the subject, we have added a comma after "verification" to clarify.

p9 L18:  add a comma after "perfect" done.

p9 L19:  add a comma after "condition" done.

p9 L25:  put the reference in () done.

p10 L2:  add a comma after "model" or remove the one after
"distributions" This entire section has been rewritten in response to other
comments.

p10 L10:  add a comma before "while" see previous comment.

p12 L18:  add a comma before "meaning" done.

p12 L31:  change "there" to "their" (or possibly "three"?) Done.

p14 Eq (7):  The capital H's should be italic here – no need to
linearize yet. done.

`p15 L27; p16 L3; p16 L10: "NKF" – do you mean "EnKF" here?` Yes, changed throughout.

`p16 L6: Capitalize to get "The biggest"` Done.

**0.2 Comment from T. Kaminski**

these comments were very helpfully included in an edited version of the LATEX article. We have copied these here in the usual font but they lack the page number context.

`doesn't the rayner time-dependent inversion study fit better than` *Chevallier et al.* `(2010) ref?` We think not. Both demonstrate the same evolution towards explicit statistical methods but the computational complexity of *Chevallier et al.* (2010) is greater.

`the bibtex entry has 2004 as the year, I am almost sure it is 2005 and have attached an update for your .bib` Fixed.

`I think it is capital P` fixed.

`and intuitive` We think this is covered in our language.

`should be the closed interval including 0 and 1, i.e.` $[0,1]$ fixed.

`and maybe better domain than space?` Agreed, fixed.

`is "her" a modern method of avoiding the male sex, like use of the plural, or just a typo?` Intentional, a counterpart to the "his" in the first part.

`maybe better density?` Agreed, changed.

`Tarantola says that his concept of states of information is more general than Bayes' theorem` This is true and we haven't followed the

approach via conditional probability but we think making the distinction explicit is an unnecessary complication here.

```
Here you use target variables, but later unknows.  To me target
variables is a bit misleading, because this is not what we are
ultimately after, you use target quantity below for that, which
I think it good.  So it might be cleare to use "unknowns" (or
"control variables") already here.
```
Reviewers have given different advice here which reflects the problem of choosing nomenclature which satisfies everyone. We think the choice of "target variables" for the things we optimise and "quantity of interest" for the output we want is a good choice.

```
maybe observations?
```
Agreed, fixed.

```
It is difficulat where to define the interface between the
model and the observation operator.  In the manuscript
on observation operators, I wrote that the domain of the
observation operator is the state space and added that it can
also be a sequence of states (for time averaged observations).
For the aggregation error, I would not blame the observational
operator (i.e.  the transport model), but the source model,
which prescibes large regions.  One could also adopt the view
point that the observation operator is not even the transport
model but the extraction of the simulated concentration at the
observational times and location, including some averaging.
```
This touches on a point made by Reviewer III. Our solution is to have the observation operator act on model state $z$ rather than target variables $x$ and to mention in Section 5.1 and throughout that the choice of target variables has important implications for the roles played by various models.

```
I find this difficult to understand
```
We have rewritten this section.

`I'd drop forward here` Agreed, changed.

`I recall what we did:  Plot Cost function and cost function with Gaussian approximation over one Q10, but have difficulty following this paragraph.` We have clarified the paragraph.

`this last sentence is difficult to understand` This refers to the paragraph on rejection sampling. We have rewritten the paragraph.

`maybe define the solution, maybe with step 1?` Done.

`what does serial correlation mean?` It seems a more common term than the synonymous autocorrelation, we have retained it.

`I have trouble following the recipe` The most important part is the reference but we have rewritten the paragraph to, hopefully, clarify.

`but I would guess there are parallel versions around that test so and so many new parameter values in parallel?` there are related methods that can do this and we have cited an example.

`here an example would have helped me` We have added a simple example.

`I am not good in the theory of differential equations.  Maybe there are relevant cases where second derivatives are involved?` We think this is covered by the phrase "usually though not necessarily"

`Could say that we mean time steps here, also to contrast the iterative procedures for solving the individual inverse problems at each time step, holds also for the particle filter described later` Added in both cases.

`don't you make a prediction for the next time step, i.e..  you need an index n+1 here` the superscript f indicates the result of the forward model. We have added this.

`or even forecast!` The forecast isn't the data assimilation step. We can think of this as the diagnostic vs prognostic parts of CCDAS.

`Could explain how a TDI fits in/relates to such a system` We don't think this is a helpful place to do this. Atmospheric inversions aren't a very natural fit to the KF since their fluxes lack an underlying dynamical model so the explanation is likely to confuse a novice.

`I think the next equation is not referred to later, so you might as well stop this sub-section here` We agree and have removed the rest of the paragraph. This is probably one of the places where reviewer I thought we veered towards being over-specific.

`maybe center?` We disagree, it is important for readers to realise not all PDFs are symmetric.

`maybe just give the number from your TDI here as example?` Even the most recent version (*Rayner et al.*, 2015) is much smaller than, say, *Chevallier et al.* (2010), the order of magnitude is probably best here.

`but this does not address the above mentioned difficulty related the high dimensional space of unknowns any better, and in low dimensional spaces with linear models one can do a Jacobian calculation instead` We believe, along with the EnKF community, that ensembles can, in fact, capture the important parts of the covariance with many fewer realisations than the rank.

`maybe more of the linearity of M and H` Both are probably important so we have added a comment on linearity.

`explain spurious correlations` Not sure what the reviewer means here. "Spurious" means unreal but that is only a tautology.

`on top of the above mentioned comment: n should be n+1, it is`

`not clear to me why you need the superscript m now` We should use f
as with the KF (changed) and apart from that the same explanation holds. We have
followed the same notation as the KF.

**References**

Chevallier, F., et al., Co2 surface fluxes at grid point scale estimated from a global 21 year
    reanalysis of atmospheric measurements, *Journal of Geophysical Research: Atmospheres*,
    *115*(D21), n/a–n/a, doi:10.1029/2010JD013887, 2010.

Evans, S. N., and P. Stark, Inverse problems as statistics, *Inverse problems*, *18*, R55–R97,
    2002.

Ide, K., P. Courtier, M. Ghil, and A. C. Lorenc, Unified notation for data assimilation: Opera-
    tional, sequential and variational, 1997.

Rayner, P. J., M. R. Raupach, M. Paget, P. Peylin, and E. Koffi, A new global gridded dataset
    of $CO_2$ emissions from fossil fuel combustion: 1: Methodology and evaluation, *J. Geophys.
    Res.*, *115*, D19,306, doi:10.1029/2009JD013439, 2010.

Rayner, P. J., A. Stavert, M. Scholze, A. Ahlström, C. E. Allison, and R. M. Law, Recent changes
    in the global and regional carbon cycle: analysis of first-order diagnostics, *Biogeosciences*,
    *12*(3), 835–844, doi:10.5194/bg-12-835-2015, 2015.

Tarantola, A., *Inverse problem theory and methods for model parameter estimation*, siam, 2005.

Tippett, M. K., J. L. Anderson, C. H. Bishop, T. M. Hamill, and J. S. Whitaker, Ensemble square
    root filters*, *Monthly Weather Review*, *131*(7), 1485–1490, 2003.

Wikle, C. K., and L. M. Berliner, A bayesian tutorial for data assimilation, *Physica D: Nonlinear
    Phenomena*, *230*(1), 1–16, 2007.

Zhang, S., X. Zheng, J. M. Chen, Z. Chen, B. Dan, X. Yi, L. Wang, and G. Wu, A global
    carbon assimilation system using a modified ensemble kalman filter, *Geoscientific Model
    Development*, *8*(3), 805–816, doi:10.5194/gmd-8-805-2015, 2015.